# On the Paradox of Certified Training

**Nikola Jovanović\***
*Department of Computer Science, ETH Zurich*

*nikola.jovanovic@inf.ethz.ch*

**Mislav Balunović\***
*Department of Computer Science, ETH Zurich*

*mislav.balunovic@inf.ethz.ch*

**Maximilian Baader**
*Department of Computer Science, ETH Zurich*

*mbaader@inf.ethz.ch*

**Martin Vechev**
*Department of Computer Science, ETH Zurich*

*martin.vechev@inf.ethz.ch*

**\*** Equal contribution

**Reviewed on OpenReview:** *https: // openreview. net/ forum? id= atJHLVyBi8*

## Abstract

Certified defenses based on convex relaxations are an established technique for training provably robust models. The key component is the choice of relaxation, varying from simple intervals to tight polyhedra. Counterintuitively, loose interval-based training often leads to higher certified robustness than what can be achieved with tighter relaxations, which is a well-known but poorly understood paradox. While recent works introduced various improvements aiming to circumvent this issue in practice, the fundamental problem of training models with high certified robustness remains unsolved. In this work, we investigate the underlying reasons behind the paradox and identify two key properties of relaxations, beyond tightness, that impact certified training dynamics: continuity and sensitivity. Our extensive experimental evaluation with a number of popular convex relaxations provides strong evidence that these factors can explain the drop in certified robustness observed for tighter relaxations. We also systematically explore modifications of existing relaxations and discover that improving unfavorable properties is challenging, as such attempts often harm other properties, revealing a complex tradeoff. Our findings represent an important first step towards understanding the intricate optimization challenges involved in certified training.

## 1 Introduction

Recent years have witnessed an increased interest in developing methods for efficiently training provably robust machine learning models. Several core techniques are based on convex relaxations (e.g., CROWN (Zhang et al., 2018), hBox (Mirman et al., 2018)), which provide robustness guarantees by approximating the effect of network layers on the input specification. A key property of a convex relaxation is its *tightness*, indicating how close it is to the non-convex shape it overapproximates.

**The Paradox of Certified Training.** As tighter relaxations are more desirable for certification (Salman et al., 2019b; Singh et al., 2019b), a natural belief is that tightness is also favorable for relaxations when used in training as part of a certified defense. Surprisingly, several prior works (Gowal et al., 2018; 2019; Zhang et al., 2020; Balunovic & Vechev, 2020; Lyu et al., 2021; Lee et al., 2021) have noticed that training with IBP (Gowal et al., 2018), which is a loose relaxation that performs poorly for certification of undefended models, allows for higher certified robustness compared to training with tighter relaxations. We illustrate this on a real model in Table 1. We easily observe a paradox: *tighter relaxations obtain worse results.* More specifically, none of the tighter relaxations can consistently outperform the loose IBP.

Table 1: The Paradox of Certified Training: training with tighter relaxations leads to worse certified robustness, failing to outperform the loose IBP relaxation. Tightness formalization and further details given in Section 3.

| Relaxation | Tightness | Certified (%) |
|---|---|---|
| IBP / Box | 0.73 | 86.8 |
| hBox / Symbolic Intervals | 1.76 | 83.7 |
| CROWN / DeepPoly | 3.36 | 70.2 |
| DeepZ / CAP / FastLin / Neurify | 3.00 | 69.8 |
| CROWN-IBP (R) | 2.15 | 75.4 |

The paradox has strongly influenced the field of certified training. While some hypothesize that it occurs due to tighter relaxations introducing difficult optimization problems (Balunovic & Vechev, 2020; Lee et al., 2021), the underlying reasons for this difficulty remain unclear. Identifying these reasons and understanding how relaxations affect training is important yet very challenging as: (i) convex relaxations require more complex (symbolic) computations than those of standard (concrete) forward passes, and thus cannot directly benefit from existing convergence results, and (ii) relaxations come with different, previously unexplored properties, and identifying precisely those which affect certified training, is difficult. In light of this, recent advances primarily focus on mitigating the practical effects of the paradox by improving the underlying optimization (Balunovic & Vechev, 2020; Zhang et al., 2020; Shi et al., 2021; Lyu et al., 2021). While these developments have advanced the state of the art, a large gap between empirical and provable robustness of models remains (Li et al., 2020; Croce et al., 2020), and we still lack principled investigations of the paradox.

**This Work.** In this work we take a step towards addressing this void and understanding the paradox of certified training. We hypothesize that two additional properties beyond tightness strongly impact certified training. First, we notice that some relaxations optimize *discontinuous* losses during training. Second, we find that some relaxations are *sensitive* to changes in weights, introducing locally non-linear loss landscapes. As they induce more complex losses, both of these properties can have negative impact on optimization, and consequently lead to low certified robustness.

While the results in Table 1 seem contradictory if considering only tightness, additionally considering continuity and sensitivity provides a more viable explanation and helps demystify the paradox. Concretely, tighter relaxations in Table 1 are harmed by discontinuity or high sensitivity of their loss, shedding light on why they do not outperform the continuous and non-sensitive IBP. On a range of datasets and architectures, our experimental evaluation further substantiates the importance of considering these two additional properties in order to gain a deeper understanding of certified training dynamics.

**Main Contributions.** Our key contributions are:

- Two fundamental properties, continuity and sensitivity, that along with tightness influence the success of a convex relaxation when used in certified training (Section 4).

- Extensive experiments on a range of convex relaxations, substantiating our hypothesis that considering continuity and sensitivity is necessary to understand the paradox of certified training (Section 5).

- A study of systematic changes to existing relaxations, showing that improving an unfavorable property of a relaxation is challenging, as this often negatively affects other properties (Section 6).

We believe the ideas presented in our work benefit further investigations of the paradox, as well as future attempts to derive new certified defenses that obtain state-of-the-art experimental results. Our paper is structured as follows. First, we provide the necessary background (Section 2) and state the paradox more formally (Section 3). In Section 4, we present our core results on continuity and sensitivity of popular relaxations. In Section 5, we provide detailed experimental evidence supporting our findings. Finally, in Section 6 we present a study of relaxation modifications, demonstrating complex dependencies between properties which complicate the process of improving unfavorable properties of existing relaxations.

## 2 Background and Related Work

We now discuss related work and provide the necessary background on training and certifying with convex relaxations. We present this background within a common framework (Salman et al., 2019b), capturing various single neuron relaxations to simplify analysis and comparison.

The discovery that neural networks are not robust to small input perturbations (Szegedy et al., 2013) led to defenses based on adversarial training (Goodfellow et al., 2015; Madry et al., 2018), hardening the model by training with adversarial examples. While adversarial defenses attain good empirical robustness, they lack robustness guarantees. Popular certification methods leverage convex relaxations (Wong & Kolter, 2018; Gehr et al., 2018; Singh et al., 2018; Raghunathan et al., 2018b; Singh et al., 2019b; Dathathri et al., 2020; Xu et al., 2020; Lyu et al., 2021), a comprehensive exposition of which can be found in Salman et al. (2019b)—here, we only provide an overview needed to understand our work. We focus on *linear relaxations*, as they are scalable (e.g., can certify ResNet34 (Serre et al., 2021)), contrary to SDP (Raghunathan et al., 2018b; Dathathri et al., 2020) which is limited to smaller networks. Other, prohibitively costly approaches, use multi-neuron relaxations (Singh et al., 2019a; Tjandraatmadja et al., 2020; Müller et al., 2021; Wang et al., 2021a), or rely on SMT (Katz et al., 2017) and MILP (Tjeng et al., 2019) solvers. Two fundamentally different competitive approaches, not our focus, are $l_\infty$-distance nets (Zhang et al., 2021), used together with relaxations, that also give rise to optimization difficulties (recently tackled in Zhang et al. (2022)), and randomized smoothing (Cohen et al., 2019; Salman et al., 2019a) which is more scalable than convex relaxations but offers only probabilistic guarantees, and introduces work at inference time, making it unsuitable for certain applications.

**Setting.** We consider an $L$-layer feedforward ReLU network $h = h_L \circ h_{L-1} \circ \cdots \circ h_1$ with parameters $\boldsymbol{\theta}$, where $h_i \colon \mathbb{R}^{n_{i-1}} \to \mathbb{R}^{n_i}$ is the transformation applied at layer $i$. Let the network input be $\boldsymbol{x}_0 \in \mathbb{R}^{n_0}$ and let $\boldsymbol{x}_i := h_i \circ \cdots \circ h_1(\boldsymbol{x}_0)$ be the result after layer $i$, for $i \in [L]$, where $[L] := \{1, \ldots, L\}$. Each $h_i$ is either a dense/convolutional layer, both of which can be viewed as an affine transformation $\boldsymbol{x}_i = \boldsymbol{W}_i \boldsymbol{x}_{i-1} + \boldsymbol{b}_i$, or a nonlinear ReLU layer $\boldsymbol{x}_i = \max(\boldsymbol{x}_{i-1}, 0)$, where max is applied componentwise. Further, we assume the two layer types alternate, with $h_1$ and $h_L$ being affine. We focus on classification, where inputs $\boldsymbol{x}_0$ are classified to one of $n_L$ classes based on the logit vector $\boldsymbol{z} \equiv \boldsymbol{x}_L$, and the case of $\ell_\infty$ robustness. Namely, to certify robust classification to label $y$ in an $\ell_\infty$ ball of radius $\epsilon > 0$ around $\boldsymbol{x}$, we prove that for every $y' \neq y$

$$\boldsymbol{c}_{y'}^T \boldsymbol{z} < 0, \quad \boldsymbol{z} := h(\boldsymbol{x}_0), \ \forall \boldsymbol{x}_0, \|\boldsymbol{x} - \boldsymbol{x}_0\|_\infty < \epsilon, \tag{1}$$

where $\boldsymbol{c}_{y'} = \boldsymbol{e}_{y'} - \boldsymbol{e}_y$, by upper bounding the left-hand side with a negative value.

Given some $\epsilon$, and a set $D$ of input examples $(\boldsymbol{x}, y) \in \mathbb{R}^{n_0} \times [n_L]$, we use $CR(\boldsymbol{\theta}, \epsilon, m) \in [0, 1]$ to denote the *certified robustness* of a network with parameters $\boldsymbol{\theta}$ under certification method $m$, i.e., the ratio of examples from $D$ for which $m$ is able to prove that the network satisfies Equation 1.

**Convex Relaxations.** On an intuitive level, convex relaxations are a class of methods for robustness certification of neural networks that attempt to prove Equation 1 by deriving and propagating bounds on possible values of intermediate results, overapproximating (i.e., *relaxing*) the effect of non-linear activations in the network to obtain a computationally efficient certificate.

More formally, certification with convex relaxations proceeds through the network layer by layer, producing elementwise lower and upper bounds of $\boldsymbol{x}_i$, $\boldsymbol{l}_i \in \mathbb{R}^{n_i}$ and $\boldsymbol{u}_i \in \mathbb{R}^{n_i}$ respectively. Starting from $\boldsymbol{l}_0 = \boldsymbol{x} - \epsilon$ and $\boldsymbol{u}_0 = \boldsymbol{x} + \epsilon$ we aim to obtain $\boldsymbol{l}_L$ and $\boldsymbol{u}_L$, which yields the desired upper bounds of $\boldsymbol{c}_{y'}^T \boldsymbol{z}$ for all $y'$, allowing us to verify the robustness property. To this end, all following methods (*linear* relaxations) maintain one upper and one lower linear bound for each neuron $x_{i,j}$ of layer $i$:

$$\underline{\boldsymbol{a}_{ij}}^T \boldsymbol{x}_{i-1} + \underline{d_{ij}} \leq x_{i,j} \leq \overline{\boldsymbol{a}_{ij}}^T \boldsymbol{x}_{i-1} + \overline{d_{ij}}, \tag{2}$$

where $\underline{\boldsymbol{a}_{ij}}, \overline{\boldsymbol{a}_{ij}} \in \mathbb{R}^{n_{i-1}}$ and $\underline{d_{ij}}, \overline{d_{ij}} \in \mathbb{R}$ for all $j \in [n_i]$. Excluding the *IBP* relaxation, all methods use $x_{i,j} = (\boldsymbol{W}_i \boldsymbol{x}_{i-1})_j + b_{i,j}$ for linear layer bounds, $x_{i,j} = 0$ if $u_{i-1,j} \leq 0$ and $x_{i,j} = x_{i-1,j}$ if $l_{i-1,j} \geq 0$ for stable ReLU bounds, and calculate $\boldsymbol{l}_i$ and $\boldsymbol{u}_i$ using *backsubstitution* introduced next. Unstable ReLU ($l_{i-1,j} < 0 < u_{i-1,j}$) bounds are method-specific and can depend on $\boldsymbol{l}_{i-1}$ and $\boldsymbol{u}_{i-1}$.

**Backsubstitution.** Starting with $x_{i,j} \leq \overline{\boldsymbol{a}_{ij}}^T \boldsymbol{x}_{i-1} + \overline{d_{ij}}$ (similarly for the lower bound), we substitute $\boldsymbol{x}_{i-1}$ by replacing each $x_{i-1,j'}$ with its respective upper bound $\overline{\boldsymbol{a}_{i-1,j'}} \boldsymbol{x}_{i-2} + \overline{d_{i-1j'}}$ if $(\overline{\boldsymbol{a}_{ij}})_{j'}$ is positive and with its

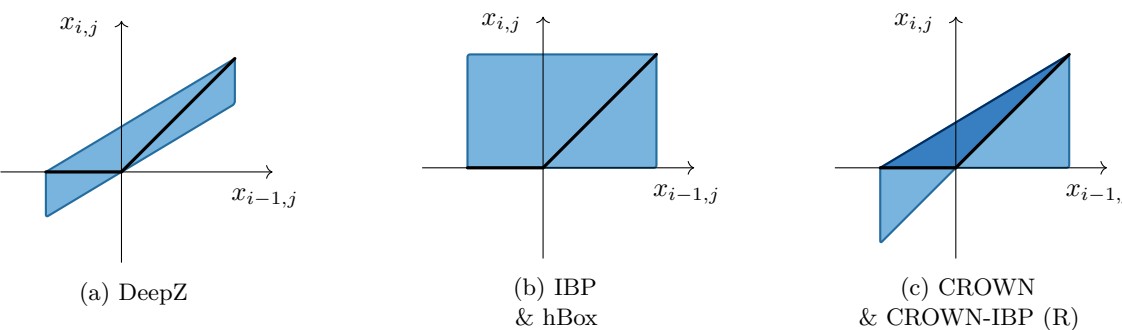

(a) DeepZ

(b) IBP
& hBox

(c) CROWN
& CROWN-IBP (R)

Figure 1: Illustration of unstable ReLU convex relaxations for methods introduced in Section 2.

lower bound otherwise. This is repeated recursively through the layers until we reach constraints of the form

$$\underline{\boldsymbol{p}_j}^T \boldsymbol{x}_0 + \underline{q_j} \le x_{i,j} \le \overline{\boldsymbol{p}_j}^T \boldsymbol{x}_0 + \overline{q_j}, \tag{3}$$

where $\underline{\boldsymbol{p}_j}^T, \overline{\boldsymbol{p}_j}^T \in \mathbb{R}^{n_0}$ and $\underline{q_j}, \overline{q_j} \in \mathbb{R}$ for all $j \in [n_i]$. Here, we can in turn substitute the appropriate side of $\boldsymbol{l}_0 \le \boldsymbol{x}_0 \le \boldsymbol{u}_0$ for each element in $\boldsymbol{x}_0$, to obtain a lower and upper bound $l_{i,j}$ and $u_{i,j}$ on $x_{i,j}$ solely w.r.t. the bounds of $\boldsymbol{x}_0$. We provide further details of this procedure in Appendix A.

Note that while some of the relaxations have more efficient implementations, they produce the same outputs as our formulation. We use this formulation as it allows us to capture all of the needed relaxations, and further, our results are conceptual and hold for any implementation.

**Tightness.** Given a network robust around $\boldsymbol{x}$, the success of certification with a relaxation depends on its *tightness*. Intuitively, tighter relaxations utilize overapproximations that are closer to the approximated non-convex shapes, and produce tighter bounds $l_{i,j}$ and $u_{i,j}$ on each $x_{i,j}$. For a small number of relaxation pairs $(r_1, r_2)$, e.g., hBox and IBP (see Appendix B.2), we can prove that $r_1$ is *strictly tighter* than $r_2$ (i.e., each bound is strictly tighter), implying that for any fixed network and perturbation, every example that is certified by $r_2$ is also certified by $r_1$. For most other pairs there is a consistent empirical understanding of relative tightness (Salman et al., 2019b; Singh et al., 2019b), which we will aim to quantify in Section 3. We now proceed to introduce the specifics of commonly used relaxations.

**DeepZ.** The *DeepZ* relaxation (Singh et al., 2018), equivalent to *CAP* (Wong & Kolter, 2018), *Fast-Lin* (Weng et al., 2018), and *Neurify* (Wang et al., 2018b), uses the following for unstable ReLUs (Figure 1a):

$$\lambda x_{i-1,j} \le x_{i,j} \le \lambda x_{i-1,j} - \lambda l_{i-1,j},$$

where $\lambda := u_{i-1,j}/(u_{i-1,j} - l_{i-1,j})$.

**IBP/hBox.** The *IBP* (Gowal et al., 2018) or *Box* (Mirman et al., 2018; Gehr et al., 2018) relaxation uses interval arithmetic instead of backsubstitution, ignoring other dependencies. For affine layers, the upper bound (similarly for the lower bound) is $(\boldsymbol{W}_i \boldsymbol{h}_{i-1})_j + b_{i,j}$ where $h_{i-1,j} = u_{i-1,j}$ if the corresponding element of $\boldsymbol{W}_i$ is positive, and $h_{i-1,j} = l_{i-1,j}$ otherwise. For ReLU, it uses $\text{ReLU}(l_{i-1,j}) \le x_{i,j} \le \text{ReLU}(u_{i-1,j})$ (Figure 1b). *hBox* is an instantiation of a *hybrid zonotope* (Mirman et al., 2018), also called *symbolic interval* in Wang et al. (2018a). It uses the same bounds as IBP, $0 \le x_{i,j} \le u_{i-1,j}$, for unstable ReLUs, replacing $x_{i,j}$ with these bounds in the rest of backsubstitution. For stable ReLUs and affine layers, as with all other methods except IBP, it uses $x_{i,j} = (\boldsymbol{W}_i \boldsymbol{x}_{i-1})_j + b_{i,j}$ and $x_{i,j} = x_{i-1,j}$ ($x_{i,j} = 0$), respectively.

**CROWN/CROWN-IBP (R).** *CROWN* (Zhang et al., 2018) and *DeepPoly* (Singh et al., 2019b) have the same upper bound as DeepZ for unstable ReLUs, but choose the lower bound adaptively: $0 \le x_{i,j}$ if $-l_{i-1,j} \ge u_{i-1,j}$, or $x_{i-1,j} \le x_{i,j}$ otherwise (Figure 1c). *CROWN-IBP (R)* (Zhang et al., 2020) is a variant which efficiently computes $l_i$ and $u_i$ using IBP at all layers except the last, which uses CROWN and performs a full backsubstitution.

**Certified Training.** While adversarial training often improves empirical robustness, *certified robustness* (ratio of inputs where we can guarantee robustness as in Equation 1) usually remains low for all relaxations, a

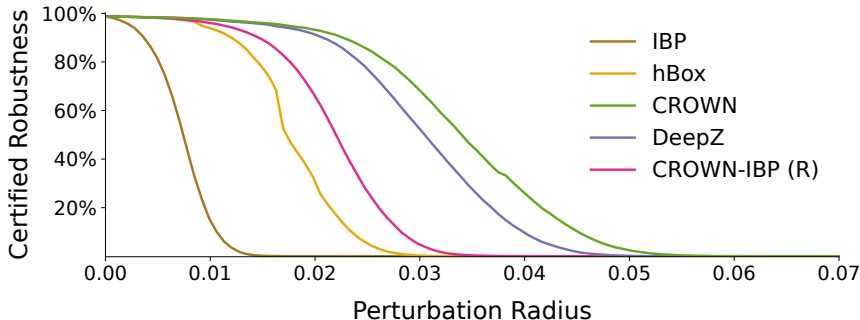

Figure 2: CR (certified robustness) curves of relaxations on a convolutional network trained on MNIST.

known observation reaffirmed in Appendix B.1. *Certified training* (Wong & Kolter, 2018; Mirman et al., 2018; Gowal et al., 2018; Raghunathan et al., 2018a; Zhang et al., 2020; Lyu et al., 2021) addresses this, aiming to produce networks amenable to certification by incorporating the certification method into training, and minimizing the cross-entropy loss $\mathcal{L}_{CE}(\hat{\boldsymbol{z}}, y)$, where $\hat{\boldsymbol{z}}$ is the worst case logit, s.t., $\hat{z}_y = l_{L,y}$ and $\hat{z}_{y'} = u_{L,y'}$ for all $y' \neq y$.

All above relaxations can be used in certified training, but certified robustness obtained this way is far from the theoretical limit—Baader et al. (2020) proved that IBP-certified networks can approximate any continuous function arbitrarily precisely. Note that in the following, we differentiate certified training with the CROWN-IBP (R) relaxation ($\beta\colon 1 \to 1$ in (Zhang et al., 2020)) from the certified defense CROWN-IBP ($\beta\colon 1 \to 0$ in (Zhang et al., 2020)) which combines CROWN-IBP (R) and IBP losses in training.

## 3   The Paradox of Certified Training

We now present *the paradox of certified training*, a well-known observation that has limited the applicability of certified defenses in practice, and discuss existing hypotheses that attempt to explain it.

**Tightness Should Help Training.** Recall from Section 2 that while we can rarely prove that a relaxation is strictly tighter than another relaxation, there is a consistent empirical understanding of their relative tightness. We illustrate this in Figure 2 by comparing *certified robustness (CR) curves* of relaxations on a fixed naturally trained MNIST network. We further quantify empirical tightness as *CR-AUC*, the area under the CR curve. While CR-AUC varies based on network choice and the training method, for a fixed setting, it can be used to compare tightness of methods, and we use it in the following when referring to tightness. As tighter relaxations certify more examples when applied to naturally trained networks, it is natural to assume that this effect extends to certified training, i.e., training with a tighter method should lead to higher CR.

**Training with Tighter Relaxations Leads to Worse Results.** Surprisingly, it is well established (Gowal et al., 2018; 2019; Balunovic & Vechev, 2020; Zhang et al., 2020; Lee et al., 2021) that this is not the case in practice, and tightness can in fact harm certified robustness when a relaxation is used in training. Most notably, it has been observed that IBP training often outperforms training with DeepZ and CROWN which are (empirically) tighter. We refer to this phenomenon as the paradox of certified training. We illustrate this paradox in Table 1, where we report CR-AUC (from the experiment in Figure 2) and certified robustness (with $\epsilon_{test} = 0.3$) after certified training of the same MNIST network with each relaxation.

**Existing Hypotheses.**   While recent state-of-the-art certified defenses based on convex relaxations (Balunovic & Vechev, 2020; Zhang et al., 2020; Shi et al., 2021; Lyu et al., 2021) focus on mitigating the paradox in practice, the fundamental reasons behind it were so far poorly understood. Some conjecture that tighter relaxations over-regularize the network (Zhang et al., 2020), yield hard optimization problems (Balunovic & Vechev, 2020), or simply state that they unexpectedly underperform (Gowal et al., 2018; 2019; Lyu et al., 2021), but they have not investigated this further. Lee et al. (2021) provide limited theoretical results that attempt to give insights into the paradox, but are unable to explain the results of most relaxations

(e.g., hBox, CROWN, CROWN-IBP (R)) as these are discontinuous (Section 4.2), thus directly violating their Lipschitz continuity assumptions.

## 4 Properties of Convex Relaxations

We investigate the reasons behind the paradox discussed in Section 3. Concretely, we introduce two key properties of relaxations, *continuity* and *sensitivity*, and use them alongside tightness, which was the main focus of prior work, to improve our understanding of the paradox.

### 4.1 Tightness of Convex Relaxations

While tightness alone cannot explain the performance differences, it still has a significant role in the final certified robustness. We highlight this with the following theorem:

**Theorem 1.** *Let $r_1$ and $r_2$ be two convex relaxations, where $r_1$ is known to be strictly tighter than $r_2$. For a network parametrized by $\boldsymbol{\theta}$ and any $\epsilon \geq 0$, it holds that $\max_{\boldsymbol{\theta}} CR(\boldsymbol{\theta}, \epsilon, r_1) \geq \max_{\boldsymbol{\theta}} CR(\boldsymbol{\theta}, \epsilon, r_2)$.*

The theorem (see the proof in Appendix B.3) tells us that with a perfect optimizer, tightness would be the *sole* performance factor of relaxations, provided that one is strictly tighter than the other. However, our results in Table 1 demonstrate that this does not happen in practice, e.g., despite hBox being strictly tighter than IBP, training with it results in worse certified robustness. Clearly, gradient-based optimization in practice leads to worse parameters for tighter relaxations and the underlying reasons are unclear.

### 4.2 Continuity of Convex Relaxations

While convex relaxations represent layer constraints as convex sets, the training loss is not necessarily convex with respect to network weights. Moreover, we observe that some relaxations create a *discontinuous* loss landscape, harming first-order optimization as gradients near the discontinuity do not provide *any* information about the function values after the discontinuity (see Section 4.4). We show that CROWN, CROWN-IBP (R), and hBox all suffer from this problem. In Section 5.1 we show that these relaxations have many discontinuities when instantiated on a realistic network, but here we focus on a minimal example to better illustrate the core issue. Note that our definition of continuity is binary and depends only on the convex relaxation, without requiring knowledge of the architecture nor the training process. There could be other, more fine-grained numerical definitions, such as counting the number of discontinuities (see for example Figure 6 or Table 4) along a certain trajectory, but these may depend on the setting and necessarily require running the training, as they cannot be computed beforehand. See Appendix I for a further discussion.

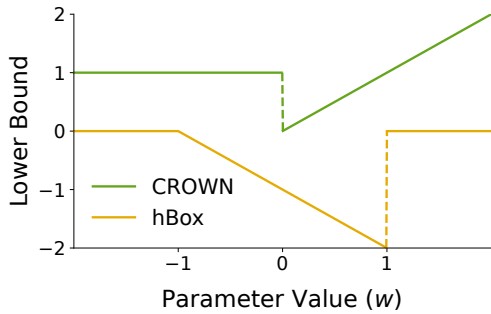

Figure 3: Discontinuity of CROWN and hBox.

We focus on the discontinuity of the output layer lower bounds $\boldsymbol{l}_L$, treating each $l_{L,j}$ as a function of the network weights. Note that all findings can be easily extended to the actual loss function $\mathcal{L}_{CE}(\hat{\boldsymbol{z}}, y)$. We construct a minimal example to produce the discontinuities: a 3-layer network with input $x_{0,1} \in [-1, 1]$, affine layer $x_{1,1} = x_{1,2} = x_{0,1} + w$ where $w$ is the only network parameter, ReLU layer $x_{2,1} = \text{ReLU}(x_{1,1})$, $x_{2,2} = \text{ReLU}(x_{1,2})$, and the output layer given as $x_{3,1} = x_{2,1} + 1$ and $x_{3,2} = x_{2,2} - x_{2,1}$ (see Appendix C.1 for an illustration of the network). Figure 3 shows the discontinuities that arise when varying the parameter $w$.

**Discontinuity of CROWN and CROWN-IBP (R).** For CROWN, the discontinuities arise due to its adaptive choice of the lower bound for unstable ReLUs (Figure 1c), used as a heuristic to tighten the bounds. In our example, assume we use CROWN to compute the lower bound $l_{3,1}$ of $x_{3,1}$. For $w \in [-1, 1]$, the ReLUs are unstable with the preactivation range $[-1 + w, 1 + w]$. Thus, for $w \in [-1, 0]$, as $-l_{2,1} \geq u_{2,1}$, CROWN picks the lower bound $x_{2,1} \geq 0$ so $l_{3,1} = 1$, and for $w \in (0, 1]$ the lower bound $x_{2,1} \geq x_{1,1}$ so $l_{3,1} = w$. This

creates a discontinuity when $-l_{2,1} = u_{2,1}$, i.e., at $w = 0$. This implies the discontinuity of CROWN-IBP (R) as it uses CROWN for its final bounds.

**Discontinuity of hBox.** The discontinuities of hBox are caused by hBox switching from simple IBP bounds (Figure 1b), to the tight relation $x_{i,j} = x_{i-1,j}$. Assume we are deriving $l_{3,2}$. For $w \in (-1, 1)$, the ReLUs are unstable, so we use IBP bounds $0 \le x_{2,j} \le u_{1,j} = 1 + w$ for $j \in \{1, 2\}$, obtaining $l_{3,2} = -1 - w$, which approaches $-2$ as $w$ approaches 1. However, for $w \ge 1$, we tighten the bound using $x_{2,j} = x_{1,j}$, resulting in $l_{3,2} = 0$, thus a discontinuity when $l_{2,1} = l_{2,2} = 0$, i.e., at $w = 1$.

As our example shows, *few neurons are sufficient to produce discontinuities.* Thus, we expect large networks to have a large number of discontinuities, appearing at any ReLU neuron $x_{i,j}$, whenever $-l_{i-1,j} = u_{i-1,j}$ (for CROWN) or $l_{i-1,j} = 0$ (for hBox). Backsubstitution accumulates this effect, creating an unfavorable landscape. As mentioned earlier, we demonstrate this in practice on a realistic network in Section 5.1.

**Continuity of Other Relaxations.** The remaining two relaxations, IBP and DeepZ, are always continuous, as formalized in the following theorem (full proof in Appendix C.2):

**Theorem 2.** *The output bounds $l_{L,j}$ of IBP and DeepZ are continuous w.r.t network parameters $\boldsymbol{\theta}$.*

*Proof Sketch.* For IBP, $\boldsymbol{l}_i$ and $\boldsymbol{u}_i$ depend only on the previous layer either linearly or via ReLU, both being continuous. For DeepZ, the key step is proving that the ReLU relaxation bounds are continuous in points where the ReLU changes stability. Recall that the unstable case bounds are $\lambda x_{i-1,j}$ and $\lambda x_{i-1,j} - \lambda l_{i-1,j}$, where $\lambda = u_{i-1,j}/(u_{i-1,j} - l_{i-1,j})$. Then, as $l_{i-1,j} \to 0$, $\lambda \to 1$ and as $u_{i-1,j} \to 0$, $\lambda \to 0$. For both, the unstable case bounds (in the limit) match the stable case ones; therefore, there is no discontinuity. $\square$

### 4.3 Sensitivity of Convex Relaxations

Next, we analyze the effect of small weight changes on the output loss by measuring the degree of change in the output when the *first* layer weights are shifted by $\delta$ in the gradient direction. While changes in other layers also matter, we consider only changes in the first layer to make the computation of the bounds tractable.

To this end, we define a set of rational functions of $\delta$ as $R_N(\delta) = \{p(\delta)/q(\delta) \mid p(\delta), q(\delta) \in P_N(\delta)\}$, where $P_N(\delta)$ denotes the polynomials of degree up to $N$. Note that $P_N(\delta) \subseteq R_N(\delta)$. We say that some neuron $x_{i,j}$ is in the set $P_N(\delta)$ (or $R_N(\delta)$) if that set contains both $l_{i,j}$ and $u_{i,j}$, now treated as functions of $\delta$ where $\delta = 0$ corresponds to the concrete $l_{i,j}$ and $u_{i,j}$ used in Section 2. Everything else is treated as a constant. During backsubstitution for $x_{i,j}$, all encountered $x_{i',j'}$ are repeatedly replaced with bound expressions from Equation 2, until we reach Equation 3 to obtain linear expressions for $l_{i,j}$ and $u_{i,j}$. If the output neurons of the network are in $R_N(\delta)$, we say that the *sensitivity* of a relaxation is $N$. Sensitivity is an undesirable property, as it introduces a complex loss landscape that hinders optimization, as further explained in Section 4.4. Note that while coefficients of the polynomials also matter, they are influenced by the weights which makes it difficult to compute the worst-case bound in closed form. In the following, we compute the sensitivity of convex relaxations (more detailed derivation is deferred to Appendix J), to show that DeepZ, CROWN and CROWN-IBP (R) are highly sensitive, while IBP and hBox are not, inducing more favorable landscapes. As before, while we focus on $\boldsymbol{l}_L$, the conclusions can be extended to the actual loss. We always consider the worst case w.r.t. all bound choices and ReLU stability, and assume all layers are of size $M$. While the sensitivity values we obtain represent an upper bound, they clearly demonstrate that some relaxations are highly sensitive, as opposed to IBP and hBox, for which our result on insensitivity is exact. This is summarized in Table 2.

**Computing the Sensitivity.** As the first layer is affine, we have $x_{1,j} \in P_1(\delta)$ for all relaxations. To compute the sensitivity, we sequentially analyze the effect of each layer.

**IBP/hBox.** For IBP, assume that at layer $i$, all $x_{i-1,j} \in P_N(\delta)$. For an affine layer, as $l_{i,j}$ and $u_{i,j}$ are linear combinations of elements of $\boldsymbol{u}_{i-1}$ and $\boldsymbol{l}_{i-1}$, we have $x_{i,j} \in P_N(\delta)$. For a ReLU layer, as $u_{i,j} = \text{ReLU}(u_{i-1,j})$ (same for $l_{i,j}$) we again have $x_{i,j} \in P_N(\delta)$. Thus, all neurons are in $P_1(\delta) \subseteq R_1(\delta)$ so the sensitivity of IBP is 1. For hBox, the only difference are affine layers, where now $x_{i,j} = (\boldsymbol{W}_i \boldsymbol{x}_{i-1})_j + b_{i,j}$. As linear combinations of elements of $P_N(\delta)$ are in $P_N(\delta)$, all neurons stay in $P_1(\delta)$ and the sensitivity of hBox is also 1.

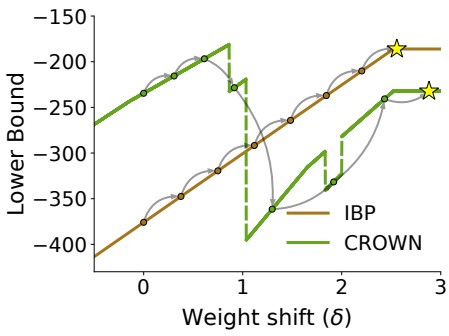 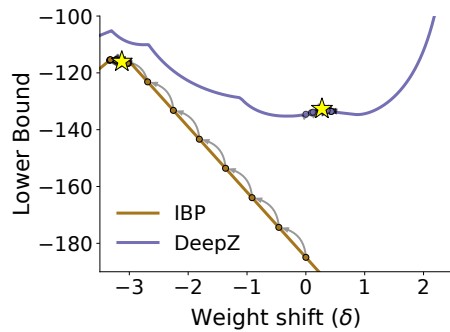

Figure 4: Convex relaxations with discontinuous (left) and highly sensitive (right) losses.

**DeepZ/CROWN.** The ReLU bounds of DeepZ, $\lambda x_{i-1,j}$ and $\lambda x_{i-1,j} - \lambda l_{i-1,j}$, significantly increase the sensitivity. After the first ReLU layer, we have that $x_{2,j} \in R_2(\delta)$ as $\lambda \in R_1(\delta)$. This changes the behavior of all following affine layers, as a linear combination of $M$ elements of $R_N(\delta)$ is in $R_{MN}(\delta)$. Thus, $x_{3,j} \in R_{2M}(\delta)$. For the following ReLU layers, if we assume the inputs are in $R_N(\delta)$, we have that $\lambda \in R_{2N}(\delta)$, and thus the outputs are in $R_{3N}(\delta)$. Putting this together, each ReLU-affine block from layer 4 onwards multiplies the sensitivity by $3M$. As there are $B \equiv \lfloor L/2 \rfloor - 1$ such blocks, we obtain $2 \cdot 3^B M^{B+1}$ for the final sensitivity. CROWN uses the same upper ReLU bounds as DeepZ, so we can apply the same analysis, and show that the sensitivity of CROWN is $2 \cdot 3^B M^{B+1}$ as well. Thus, both DeepZ and CROWN are highly sensitive.

**CROWN-IBP (R).** Here, at ReLU layer $i$ during the (only) backsubstitution, we have to consider $l_{i-1,j}$ and $u_{i-1,j}$ separately from $x_{i-1,j}$. While the former were precomputed with IBP, and are thus in $P_1(\delta)$, the latter get substituted as usual and can carry larger sensitivity. Assuming $x_{i-1,j} \in R_N(\delta)$ $(N \geq 1)$ and observing that we always have $\lambda \in R_1(\delta)$, it follows that $x_{i,j} \in R_{N+1}(\delta)$. As the affine layers have the same effect as before, each ReLU-affine block now increases sensitivity from $N$ to $(N+1)M$. As before, $x_{3,j} \in R_{2M}(\delta)$, so summing the arising geometric series gives the final sensitivity of $\frac{2M^{B+2} - M^{B+1} - M}{M-1}$, which is in $\mathcal{O}(M^{B+1})$. Clearly, CROWN-IBP (R) is also significantly more sensitive than IBP and hBox.

## 4.4 Continuity and Sensitivity Impact Optimization

We now discuss how discontinuity and high sensitivity negatively affect optimization with gradient descent (GD). We consider a randomly initialized network and optimize first layer weights via GD, trying to *maximize* the lower bound of one output neuron, produced using a particular relaxation. For plotting, we restrict the optimization to the direction of the gradient in the initial point $\delta = 0$ (see Appendix D for details).

**The Impact of Discontinuities.** The key issue with discontinuous relaxations is that GD can, at a discontinuity, fall off a cliff in the landscape to a region from which it fails to recover—i.e., where gradients lead it to a suboptimal local maximum. Figure 4 (left) shows a manifestation of this issue. Even though the landscape of CROWN allows for a higher solution than IBP, GD with CROWN converges to a worse value than IBP. Contrary to this, continuous relaxations such as IBP allow GD to easily navigate the landscape. This matches the literature stating that optimizing discontinuous functions requires complex algorithms (Conn & Mongeau, 1998; Martínez, 2002; Wechsung & Barton, 2014).

**The Impact of High Sensitivity.** Sensitive relaxations introduce a complex loss landscape with a larger number of local optima and saddle points, where GD can get stuck. Figure 4 (right) is an example where DeepZ has a highly non-linear landscape that traps GD at a local maximum with a low objective value, not allowing it to progress to better solutions. While DeepZ is tighter for all $\delta$, IBP has the minimum sensitivity and is thus piecewise linear, allowing GD to quickly converge to a higher value. Extensive theory (Pardalos & Vavasis, 1991; Jibetean & de Klerk, 2006; Zoej et al., 2007) confirms that high-degree polynomial and rational functions, which appear for sensitive relaxations, are hard to optimize.

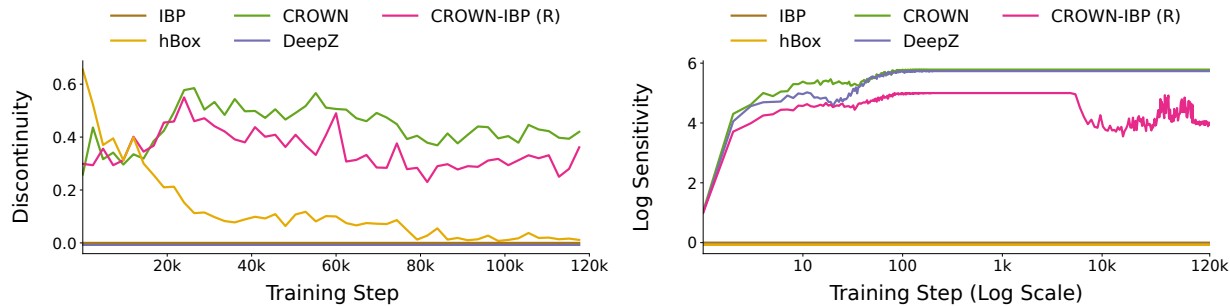

Figure 6: Discontinuity (left) and sensitivity (right) during certified training with different relaxations.

## 5 Experimental Evaluation

In this section we perform an experimental evaluation, further substantiating our hypothesis on the properties of relaxations that explain the paradox of certified training. First, in Section 5.1 we show that discontinuities and high sensitivity appear in practice. Then, in Section 5.2, we provide a deeper insight into the paradox by evaluating certified training and confirming our claims regarding the effect of continuity and sensitivity.

### 5.1 Continuity and Sensitivity in Practice

First, we measure continuity and sensitivity on a naturally trained network. Namely, we train FC, a 5-layer network (see Appendix F.1) on MNIST. Then, we measure the change in the lower bound of one output neuron as we shift all first layer weights by $\delta$ in the gradient direction of that neuron. In Figure 5, we show the resulting bounds depending on $\delta$, on a representative input with $\epsilon = 0.15$. Similar results are observed for different choices of $\delta$ and $\epsilon$. The experiment confirms our results: hBox, CROWN, and CROWN-IBP (R) indeed suffer from discontinuities, while IBP and DeepZ do not. We observe that CROWN has more discontinuities than other relaxations due to its adaptive lower bound. We further confirm our findings on more networks in Appendix E.1.

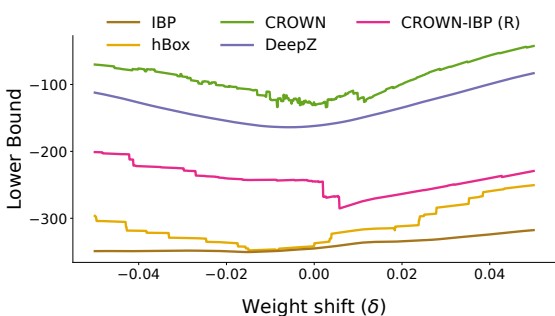

Figure 5: Continuity and sensitivity on a naturally trained MNIST network.

Additionally, in Figure 6 we measure continuity and sensitivity during certified training for each convex relaxation (complete details of the experiment provided in Appendix E.2). We can observe (left) that hBox is discontinuous at the start of training when more ReLUs are changing stability, and becomes more continuous as they stabilize, when CROWN is more discontinuous due to a larger percentage of consistently unstable ReLUs. These observations match our results on continuity from Section 4.2. While DeepZ is continuous, we can notice (right) that it is highly sensitive already very early in training, which explains its bad performance when used in certified training, contrary to what might be expected given its favorable tightness.

### 5.2 Evaluation of Certified Training

Next, we perform a thorough evaluation of certified training with all relaxations introduced in Section 2 on 4 widely used datasets (MNIST, FashionMNIST, SVHN, CIFAR-10) and 2 architectures: FC, a 5-layer dense network, and CONV, a 3-layer convolutional network. For CIFAR-10 we use the larger 4-layer CONV+, here necessary to obtain nontrivial accuracies after certified training. Note that further increasing network size only marginally boosts the results (Zhang et al., 2020), but prevents training with time and memory intensive CROWN, which can already not be trained on CONV+. We focus on well-established and challenging cases

Table 2: Certified robustness (in %) after certified training with convex relaxations. Symbol × indicates that a property is unfavorable, i.e. the relaxation has a discontinuous loss landscape or is highly sensitive. We use $\epsilon_{test} = 0.3$ for MNIST and FashionMNIST, and $\epsilon_{test} = 8/255$ for SVHN and CIFAR-10 datasets.

| Method | Continuity | Sensitivity | MNIST | | FashionMNIST | | SVHN | CIFAR-10 |
| | | | FC | CONV | FC | CONV | CONV | CONV+ |
|---|---|---|---|---|---|---|---|---|
| IBP | ✓ | ✓ | **74.0** | **86.8** | 40.4 | **52.0** | **28.9** | **29.0** |
| hBox | × | ✓ | 57.0 | 83.7 | 39.6 | 47.1 | 23.6 | 20.0 |
| CROWN | × | × | 57.3 | 70.2 | 30.2 | 31.5 | 23.4 | OOM |
| DeepZ | ✓ | × | 64.2 | 69.8 | 35.0 | 34.0 | 24.5 | 22.8 |
| CROWN-IBP (R) | × | × | 70.5 | 75.4 | **41.1** | 40.0 | 27.5 | 24.3 |

Table 3: Certified training of two MNIST networks with modifications of relaxations aimed at improving unfavorable properties. The first row shows the favorability of tightness (T), continuity (C) and sensitivity (S) for each relaxation. The following rows show certified robustness (in %).

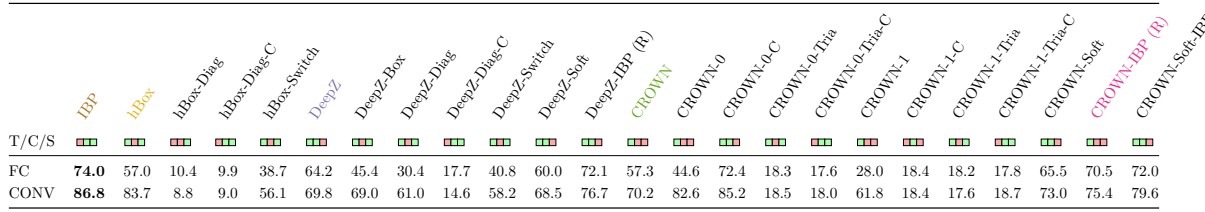

| | IBP | hBox | hBox-Diag | hBox-Diag-C | hBox-Switch | DeepZ | DeepZ-Box | DeepZ-Diag | DeepZ-Diag-C | DeepZ-Switch | DeepZ-Soft | DeepZ-IBP (R) | CROWN | CROWN-0 | CROWN-0-C | CROWN-0-Tria | CROWN-0-Tria-C | CROWN-1 | CROWN-1-C | CROWN-1-Tria | CROWN-1-Tria-C | CROWN-Soft | CROWN-IBP (R) | CROWN-Soft-IBP |
|---|---|---|---|---|---|---|---|---|---|---|---|---|---|---|---|---|---|---|---|---|---|---|---|---|
| FC | **74.0** | 57.0 | 10.4 | 9.9 | 38.7 | 64.2 | 45.4 | 30.4 | 17.7 | 40.8 | 60.0 | 72.1 | 57.3 | 44.6 | 72.4 | 18.3 | 17.6 | 28.0 | 18.4 | 18.2 | 17.8 | 65.5 | 70.5 | 72.0 |
| CONV | **86.8** | 83.7 | 8.8 | 9.0 | 56.1 | 69.8 | 69.0 | 61.0 | 14.6 | 58.2 | 68.5 | 76.7 | 70.2 | 82.6 | 85.2 | 18.5 | 18.0 | 61.8 | 18.4 | 17.6 | 18.7 | 73.0 | 75.4 | 79.6 |

of strong adversaries ([Madry et al., 2018](#); [Croce et al., 2020](#)), i.e., $\epsilon_{test} = 0.3$ for MNIST/FashionMNIST and $\epsilon_{test} = 8/255$ for SVHN/CIFAR-10. We use the same relaxation for training and certification, as this is usually optimal (see Appendix F.3). Further experimental details are provided in Appendix F.1.

**Reproducing the Paradox.** Our main results are shown in Table 2. Whereas prior work provides certain evidence, our comprehensive experiments over 5 relaxations, 4 datasets and several networks, confirm that the well-known paradox of certified training generally holds: *tighter relaxations obtain worse results*, and no tight relaxation can consistently outperform the loose IBP. Note that we aim to understand the behavior of certified training with a single relaxation. As previously noted, the paradox can in some cases be circumvented with advanced training schemes, e.g., the hybrid CROWN-IBP defense can often outperform IBP by combining CROWN-IBP(R) and IBP relaxations in training (see Appendix F.4 for expanded results).

**Understanding the Paradox.** We use × to highlight cases when one of our two key properties, continuity and sensitivity, is unfavorable for training (discontinuous loss, high sensitivity), and ✓ when it is favorable. Considering tightness as a sole property of a relaxation led to a seemingly contradictory conclusion. Once we complement tightness with our two properties, the results are less puzzling, as we can explain the inferior performance of each method compared to IBP. As discontinuity and sensitivity manifest for realistic networks (Section 5.1), and can have negative effect on gradient descent (Section 4.4), we can now expect that discontinuous and sensitive relaxations will not produce satisfactory results. This is confirmed in Table 2.

Namely, we can attribute the poor results of hBox and CROWN-IBP (R) to the discontinuities in their loss which harm gradient descent. While DeepZ is continuous, it is highly sensitive which again poses a difficulty for optimization and hurts the results. Notably, CROWN suffers from both issues, thus failing despite its tightness. We see that IBP, while loose, has favorable continuity and sensitivity, and achieves the best results. This provides novel insights on the paradox—*relaxations with unfavorable properties get worse results*.

**Excluding Alternative Explanations.** Exactly quantifying the impact of each property (tightness, continuity, and sensitivity) on the result is challenging, as it might heavily depend on the setting, e.g., dataset or network (see discussion in Section 6). However, to exclude the possibility that our conclusions are an artifact of a specific setting (e.g., they hold only for a particular weight initialization), we repeat a subset of

Table 4: Mean and standard deviation (4 random seeds) of certified robustness after certified training of the CONV network on MNIST dataset (as in Table 2) using loose parametrized relaxations described in Section 6, illustrating that the tradeoff between properties, namely continuity and tightness, can change across settings.

| Looseness Parameter $\omega$ | 0.01 | 0.5 | 1 | 1.5 | 2 | 3 | 5 |
|---|---|---|---|---|---|---|---|
| LooseIBP-C | $85.4 \pm 0.5$ | $82.0 \pm 0.6$ | $80.0 \pm 0.6$ | $77.3 \pm 0.6$ | $73.4 \pm 2.0$ | $45.4 \pm 12.2$ | $13.4 \pm 0.9$ |
| LooseIBP-DC$_1$ | $\mathbf{85.8} \pm 0.6$ | $\mathbf{83.1} \pm 0.4$ | $\mathbf{81.8} \pm 0.4$ | $\mathbf{79.6} \pm 0.4$ | $\mathbf{78.3} \pm 0.8$ | $\mathbf{73.8} \pm 0.6$ | $\mathbf{21.1} \pm 19.1$ |
| LooseIBP-DC$_{10}$ | $\mathbf{85.8} \pm 0.3$ | $\mathbf{82.5} \pm 0.7$ | $\mathbf{80.8} \pm 0.5$ | $77.6 \pm 1.3$ | $\mathbf{75.6} \pm 1.3$ | $32.7 \pm 22.1$ | $11.3 \pm 0.0$ |
| LooseIBP-DC$_{100}$ | $\mathbf{86.1} \pm 0.5$ | $82.0 \pm 0.5$ | $79.9 \pm 0.5$ | $76.7 \pm 1.0$ | $68.0 \pm 2.5$ | $20.1 \pm 15.2$ | $11.3 \pm 0.0$ |
| LooseIBP-DC$_{300}$ | $\mathbf{86.2} \pm 0.4$ | $\mathbf{82.3} \pm 0.2$ | $79.0 \pm 0.2$ | $71.9 \pm 1.2$ | $24.9 \pm 13.8$ | $11.3 \pm 0.0$ | $11.3 \pm 0.0$ |
| LooseIBP-DC$_{1000}$ | $\mathbf{86.1} \pm 0.3$ | $81.9 \pm 0.3$ | $57.0 \pm 12.9$ | $11.3 \pm 0.0$ | $11.3 \pm 0.0$ | $11.3 \pm 0.0$ | $11.3 \pm 0.0$ |

our experiments for a wider range of parameter choices, including various initializations, regularization norms, learning rates, optimizers, as well as training on subsets of the data. In all considered settings we reach the same conclusions as in Table 2, further strengthening our insights. The detailed results are in Appendix G.

## 6 Improving Unfavorable Properties of Relaxations

Given our previous results that demonstrate the negative effect of unfavorable properties on certified training, a natural follow-up question is: *can we simply improve the unfavorable properties of a relaxation to make it more successful in certified training?* To investigate this question we systematically explore and evaluate modifications of previously considered relaxations, and demonstrate that this does not immediately lead to better results, as such changes often harm other properties, inducing a complex tradeoff.

**Discovering Modifications.** To obtain the modifications, we generate all combinations of suitable choices for lower and upper linear bounds (as in Equation 2) for all three ReLU stability cases, filtering out unsound candidates (i.e., those that do not properly overapproximate ReLU), and those for which there is a strictly more favorable relaxation (i.e., provably strictly tighter and not worse in continuity and sensitivity). Additionally, we include several relaxations obtained via (i) discretely *switching* between bound choices for unstable ReLU based on a CROWN-inspired heuristic; (ii) changing the same bounds in a *soft* way, eliminating the discontinuity introduced by the switching heuristic, and (iii) computing the intermediate bounds using IBP to reduce sensitivity, as in CROWN-IBP (R).

**Properties are Entangled.** The resulting relaxations are shown in Table 3. We interpret each relaxation as a modification of one of the relaxations from Table 2 aimed at improving one property, and name them accordingly. We show the favorability of each property, and certified robustness after certified training of FC and CONV networks. Our claims on tightness of these modifications are based on empirical CR-AUC measurements in the same setting as in Figure 2 (see Appendix H for details). We defer complete descriptions of each modification, including the intended as well as unintended effects on properties, to Appendix H.

The main observation from Table 3 is that properties are not independent—*modifying a relaxation to improve a property often negatively affects another one.* For example, by fixing the lower bound $x_{i-1,j} \leq x_{i,j}$ for unstable ReLUs in CROWN ▭ to eliminate the discontinuities due to heuristic switching, we obtain CROWN-1 ▭ , which introduces a new kind of discontinuities at $u_{i-1,j} = 0$ and is slightly looser. Further, using $x_{i-1,j} \leq x_{i,j}$ for the negative case creates CROWN-1-C ▭ , which is now continuous, but significantly looser. Both of these relaxations perform worse than CROWN, while some similar changes result in improvements, implying a complex tradeoff between properties, where they differently affect the certified training in different scenarios, as previously observed in Section 5.2.

Crucially, no modification is able to outperform IBP, strengthening the conclusion that modifying existing relaxations to improve unfavorable properties is not simple and does not directly lead to state-of-the-art results. Note that the same phenomenon affects most prior work in designing convex relaxations, where tightness was the sole focus in relaxation design, which unknowingly harmed other two properties and caused bad results in certified training, leading to the paradox which we focus on in this work.

**Towards Understanding the Tradeoff.** To further demonstrate that properties can affect training in different ways for different settings, as opposed to previously explored *discrete* modifications, we consider two *parametrized* modifications of IBP—continuous *LooseIBP-C($\omega$)*, which before every ReLU layer replaces the interval arithmetic bounds $[l, u]$ with $[l - \omega, u + \omega]$, and discontinuous *LooseIBP-DC$_F$($\omega$)*, which uses $[l - \omega(\lceil F * l \rceil - F * l), u + \omega(F * u - \lfloor F * u \rfloor)]$, and is further parametrized by $F$, where larger $F$ leads to more discontinuities. Increasing the *looseness parameter* $\omega$ reduces the tightness of all relaxations. For fixed $\omega$, all *LooseIBP-DC$_F$($\omega$)* have intuitively comparable tightness, and are all strictly tighter than *LooseIBP-C($\omega$)*.

In Table 4 we present the certified training results for various values of $\omega$ and $F$, with the CONV network trained on the MNIST dataset, following the setting of Table 2. We perform 4 runs with different random seeds, and report the mean and the standard deviation. As all considered relaxations are strictly looser than IBP for any fixed $\omega$, we do not expect improvements over IBP (which achieves 86.8% CR in this case, see Table 2). However, it is interesting to determine when sacrificing the continuity of *LooseIBP-C($\omega$)* for tightness of *LooseIBP-DC$_F$($\omega$)* is beneficial (we highlight such cases for each $\omega$ in bold). Namely, we can see that in tight regimes (low $\omega$), the advantage of tightness on average outweighs the harm of discontinuities, and we get comparable or slightly more favorable results for most $F$. As we move to looser regimes (high $\omega$), the differences between relaxations become more pronounced, and discontinuities become more important relative to tightness, i.e., increasing tightness improves results only if the cost paid in discontinuities is not too large. Even taking into account the standard deviation of results for small $\omega$, where some of the relaxations are comparable, our conclusion still stands. Note that for high $\omega$, relaxations sometimes fully diverge in training, dropping CR to trivial 11.3%, which explains cells with unusually high standard deviation.

This illustrates that while the three properties of a relaxation are indicative of its performance, the underlying tradeoff can vary, and estimating the exact effects of each property in a particular setting is challenging.

## 7    Conclusions, Discussion and Future Work

Our theoretical (Section 4) and experimental (Section 5) results demonstrated that attempts to use tighter relaxations in certified training have lead to unfavorable properties such as discontinuity and high sensitivity of the loss. These novel insights on the failure of these relaxations to outperform the loose IBP represent a first step towards deeper understanding of this phenomenon. As the difference between the best empirical and certified robustness is more than 25% based on current leaderboards (Li et al., 2020; Croce et al., 2020) on CIFAR-10, we now provide a brief outlook, in light of new evidence, on possible techniques that could help close this gap, and identify several high-level directions that could be explored in future work.

**New Relaxations with Favorable Properties.** First, one could try to design a novel relaxation that is tight, and has both favorable continuity and favorable sensitivity. The results of our follow-up study in Section 6 indicate that this might be difficult, as trying to improve a property of an existing relaxation often negatively affects other properties, inducing a tradeoff with complex effects on training. Nevertheless, a relaxation with all favorable properties could still exist, as for instance Lyu et al. (2021) have obtained competitive results by using a new relaxation, and such search could be further guided by our findings.

**New Training Methods for Existing Relaxations.** Second, one could attempt to utilize existing convex relaxations in certified training with a modified training procedure, which is designed to exploit the benefits of each relaxation. Examples of this in recent work include searching for counterexamples (Balunovic & Vechev, 2020), combining several relaxations (Zhang et al., 2020) or using better initialization (Shi et al., 2021), and have shown to be a promising way to obtain state-of-the-art certified robustness. Future work could attempt to explicitly incorporate the notions of continuity and sensitivity when designing such a training procedure.

**Going Beyond Convex Relaxations.** Finally, under the assumption that the tradeoff between tightness and other properties represents a fundamental obstacle for convex relaxations, a promising possibility could be to move away from training with convex relaxations altogether and adopt a fundamentally different approach. Recently, alternative methods based on the technique of randomized smoothing (Cohen et al., 2019; Salman et al., 2019a; Yang et al., 2020) or new certification-friendly model architectures (Zhang et al., 2021; 2022) have achieved strong results, which may suggest that this avenue is the most promising. However, these methods come with their own set of challenges and tradeoffs, previously discussed in Section 2.

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

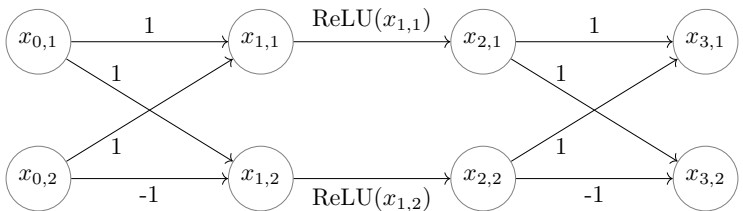

Figure 7: Toy network (from Singh et al. (2019b)).

# A  Backsubstitution Example

Here we illustrate the process of backsubstitution concretely on the toy network shown in Figure 7, using DeepZ. The same example but with the CROWN/DeepPoly relaxation is shown in (Singh et al., 2019b).

The components of the input $x_0$, namely $x_{0,1}$ and $x_{0,2}$, have bounds $-1$ and $1$, meaning that $-1 \leq x_{0,1}, x_{0,2} \leq 1$. Because the first layer $h_1$ is an affine layer we have

$$x_{1,1} = x_{0,1} + x_{0,2}, \tag{4}$$
$$x_{1,2} = x_{0,1} - x_{0,2}. \tag{5}$$

To obtain the bound $l_{1,1}$ we replace both $x_{0,1}$ and $x_{0,2}$ with $l_{0,1} = -1$ and $l_{0,2} = -1$ as their signs are both positive in Equation 4 and obtain $l_{1,1} = -2$. To obtain $l_{1,2}$ we replace $x_{0,1}$ with $l_{0,1} = -1$ and $x_{0,2}$ with $u_{0,1} = 1$ in Equation 5 as the sign of $x_{0,2}$ is negative and get $l_{1,2} = -2$. Similarly we get $u_{1,1} = 2$ and $u_{1,2} = 2$.

The second layer $h_2$ is a ReLU layer. As both ReLUs are unstable, we need to calculate $\lambda$ for each of them. As the bounds are equal, $l_{1,1} = l_{1,2} = -2$ and $u_{1,1} = u_{1,2} = 2$, we get that $\lambda$ is also equal, $\lambda = \frac{1}{2}$. Using the formula for unstable ReLUs we get

$$\frac{1}{2}x_{1,1} \leq x_{2,1} \leq \frac{1}{2}x_{1,1} + 1,$$
$$\frac{1}{2}x_{1,2} \leq x_{2,2} \leq \frac{1}{2}x_{1,2} + 1.$$

Now backsubstituting the bounds for $x_{0,1}$ and $x_{0,2}$ gives

$$\frac{1}{2}(x_{0,1} + x_{0,2}) \leq x_{2,1} \leq \frac{1}{2}(x_{0,1} + x_{0,2}) + 1,$$
$$\frac{1}{2}(x_{0,1} - x_{0,2}) \leq x_{2,2} \leq \frac{1}{2}(x_{0,1} - x_{0,2}) + 1.$$

The lower and upper bounds $l_{2,1}, u_{2,1}$ and $l_{2,2}, u_{2,2}$ for $x_{2,1}$ and $x_{2,2}$ respectively follow immediately:

$$x_{2,1} \geq \frac{1}{2}(l_{0,1} + l_{0,2}) = -1 = l_{2,1},$$
$$x_{2,1} \leq \frac{1}{2}(u_{0,1} + u_{0,2}) = 1 = u_{2,1},$$
$$x_{2,2} \geq \frac{1}{2}(l_{0,1} - u_{0,2}) = -1 = l_{2,2},$$
$$x_{2,2} \leq \frac{1}{2}(u_{0,1} - l_{0,2}) = 1 = u_{2,2}.$$

The third layer $h_3$ is again an affine layer, hence we get $x_{3,1} = x_{2,1} + x_{2,2}$ and $x_{3,2} = x_{2,1} - x_{2,2}$. In the backsubstitution step, we replace $x_{2,1}$ and $x_{2,2}$ with their upper and lower bounds and arrive at

$$x_{0,1} \leq x_{3,1} \leq x_{0,1} + 2,$$
$$x_{0,2} - 1 \leq x_{3,2} \leq x_{0,2} + 1.$$

Again, the lower and upper bounds $l_{3,1}, u_{3,1}$ and $l_{3,2}, u_{3,2}$ for $x_{3,1}$ and $x_{3,2}$ respectively follow:

$$x_{3,1} \geq l_{0,1} \qquad = -1 = l_{3,1},$$
$$x_{3,1} \leq u_{0,1} + 2 = \quad 3 = u_{3,1},$$
$$x_{3,2} \geq l_{0,2} - 1 \ = -2 = l_{3,2},$$
$$x_{3,2} \leq u_{0,2} + 1 = \quad 2 = u_{3,2}.$$

Table 5: The summary of our tightness experiments. Acc denotes standard accuracy (in %). ER denotes empirical robustness (in %), evaluated using PGD with the same $\epsilon$ used in training.

| ID | Dataset | Network | Training | Acc | ER |
|----|---------|---------|----------|-----|-----|
| A | MNIST | CONV | Natural | 98.7 | / |
| B | MNIST | CONV | PGD $\epsilon = 0.1$ | 99.0 | 94.4 |
| C | MNIST | CONV | PGD $\epsilon = 0.3$ | 98.2 | 91.0 |
| D | FashionMNIST | CONV | Natural | 91.4 | / |
| E | CIFAR-10 | CONV+ | Natural | 70.0 | / |
| F | MNIST | FC-S | Natural | 98.2 | / |
| G | MNIST | FC-S | PGD $\epsilon = 0.1$ | 99.0 | 91.8 |
| H | MNIST | FC-S | PGD $\epsilon = 0.3$ | 92.3 | 78.3 |

Table 6: The results of tightness experiments, showing CR-AUC of each method.

| Method | A | B | C | D | E | F | G | H |
|--------|-----|-----|------|------|------|------|------|-------|
| IBP | 0.73 | 0.91 | 1.94 | 0.20 | 0.02 | 0.47 | 0.63 | 2.22 |
| hBox | 1.76 | 4.76 | 10.45 | 0.53 | 0.09 | 1.42 | 2.70 | 11.44 |
| CROWN | 3.36 | 9.77 | 22.75 | 1.19 | 0.28 | 2.86 | 6.74 | 22.63 |
| DeepZ | 3.00 | 9.21 | 20.97 | 1.11 | 0.25 | 2.63 | 6.21 | 21.57 |
| CROWN-IBP (R) | 2.15 | 4.20 | 8.85 | 0.70 | 0.04 | 1.50 | 2.73 | 13.55 |

## B Additional Results on Tightness

Here we present results related to tightness, namely our experiments measuring empirical tightness of relaxations, and two proofs omitted from the main paper.

### B.1 Quantifying the Tightness of Relaxations

We present the complete results of our tightness experiments, one of which (experiment A) was shown in Table 1 and further in Figure 2.

**Setup.** We conduct 8 experiments (labeled A-H), varying the dataset (MNIST, FashionMNIST, CIFAR-10), the network (CONV and CONV+, described in Appendix F.1, and FC-S, a 100-100-10 fully-connected network), and the training method (natural training, adversarial training with PGD with $\epsilon = 0.1$, and adversarial training with PGD with $\epsilon = 0.3$). For PGD, we use 100 steps with step size 0.01. We train all models for 200 epochs. For PGD with $\epsilon = 0.3$, as necessary for convergence, we use the first 10 epochs as warm-up (natural training), and the following 50 as ramp-up, where we slowly increase the perturbation radius from 0 to $\epsilon$. We use L2 regularization with strength 5e-3 for experiment E, and 5e-5 for other experiments. The experiments are summarized in Table 5.

After training the networks, we sample 100 $\epsilon$ values from 0 to 0.07 for natural, 0 to 0.15 for PGD $\epsilon = 0.1$, and 0 to 0.4 for PGD $\epsilon = 0.3$ models. For every radius $\epsilon$, we use each relaxation to attempt to certify the examples from the test set. We calculate the certified robustness of each network under all relaxations, and plot the resulting CR curves. The CR curves shown in Figure 2 correspond to the results of experiment A. Further, we quantify the tightness of each relaxation as CR-AUC, the area under the CR curve, calculated using the trapezoidal rule. Table 1 contains CR-AUC values of curves from Figure 2 (experiment A), and certified robustness after certified training of the same network with $\epsilon_{train} = 0.3$ (part of results in Section 5.2).

**Discussion.** As the curves are similar among experiments, we summarize the results (CR-AUC) in Table 6. From these results, we can confirm two claims given in the main paper:

*It is necessary to use certified training to obtain a certifiably robust network.* We can see that adversarial training, along with improving empirical robustness, also has a positive effect on certified robustness. However, these results are significantly worse from those that can be obtained using certified training. To illustrate this point, for experiment C (MNIST, CONV, PGD with $\epsilon = 0.3$), the method that certifies the most at $\epsilon = 0.3$ is CROWN, with 11.9% certified robustness (not visible from the results shown here). Using certified training in the same setting, *all* relaxations obtain significantly better results: from 69.8% (worst method) to 86.8% (best method), as seen in Table 2.

*The relative tightness of relaxations is well established and can be empirically confirmed.* While individual CR-AUC values may vary between settings, the conclusions are consistent across all experiments.

It is worth noting that while we cover all commonly used convex relaxations, extending the tightness discussion to more complex relaxation-based methods that do not necessarily fit the framework presented in Section 2 can be challenging. As an example, recently introduced $\beta$-CROWN (Wang et al., 2021b) can in practice outperform the provably tighter Triangle (Ehlers, 2017) relaxation in certification of naturally trained networks, which may seem contradictory. However, this holds in a setup in which the former optimizes the slopes for all bounds including the intermediate layer ones, while the latter uses fixed intermediate layer bounds. For the same set of fixed intermediate bounds, as we would expect, $\beta$-CROWN can not produce better final bounds than Triangle (as stated in Corollary 3.2.1. in (Wang et al., 2021b) and previously in (Salman et al., 2019b)).

### B.2 hBox is Strictly Tighter Than IBP

Here we sketch a proof that for any neural network parameters hBox certifies more than IBP. For a formal proof see Wang et al. (2018a) or Mirman et al. (2018).

**Theorem 3** (Informal). *Given a neural network architecture parametrized by $\boldsymbol{\theta} \in \mathbb{R}^d$, for any choice of $\boldsymbol{\theta}$, hBox can certify robust classification for more inputs than IBP.*

*Proof.* To prove the statement, it is sufficient to show that the constraints introduced for each layer for hBox are tighter than IBP constraints (see Equation 3 in (Wang et al., 2018a) for more details on this claim). This is relatively straightforward for affine layers, so here we focus on ReLU layers. For unstable ReLU both use the same constraints: $0 \leq x_{i,j} \leq u_{i-1,j}$. When $u_{i-1,j} < 0$, both set $x_{i,j} = 0$. The only difference is when $l_{i-1,j} > 0$: in this case IBP uses $l_{i-1,j} \leq x_{i,j} \leq u_{i-1,j}$, while hBox sets $x_{i,j} = x_{i-1,j}$. By definition we have $l_{i-1,j} \leq x_{i-1,j} \leq u_{i-1,j}$, implying that hBox constraints are indeed tighter. $\square$

### B.3 Strictly Tighter Relaxations have Better Certified Robustness Optima

Here we restate and prove Theorem 1.

**Theorem 1.** *Let $r_1$ and $r_2$ be two convex relaxations, where $r_1$ is known to be strictly tighter than $r_2$. For a network parametrized by $\boldsymbol{\theta}$ and any $\epsilon \geq 0$, it holds that $\max_{\boldsymbol{\theta}} CR(\boldsymbol{\theta}, \epsilon, r_1) \geq \max_{\boldsymbol{\theta}} CR(\boldsymbol{\theta}, \epsilon, r_2)$.*

*Proof.* Let $\boldsymbol{\theta}_1 := \arg\max_{\boldsymbol{\theta}} CR(\boldsymbol{\theta}, \epsilon, r_1)$ and analogously $\boldsymbol{\theta}_2 := \arg\max_{\boldsymbol{\theta}} CR(\boldsymbol{\theta}, \epsilon, r_2)$. We can observe that:

$$\max_{\boldsymbol{\theta}} CR(\boldsymbol{\theta}, \epsilon, r_2) = CR(\boldsymbol{\theta}_2, \epsilon, r_2)$$
$$\leq CR(\boldsymbol{\theta}_2, \epsilon, r_1)$$
$$\leq CR(\boldsymbol{\theta}_1, \epsilon, r_1).$$

Here, the first inequality follows from the fact that $r_1$ is a strictly tighter relaxation than $r_1$ for any parameter choice. The second inequality follows from the definition of $\theta_2$ as maximum, completing the proof. $\square$

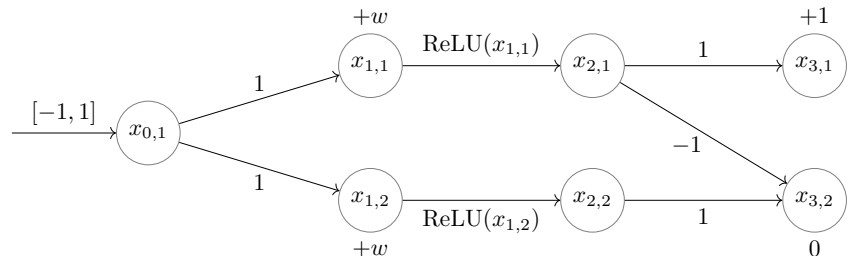

Figure 8: The network used in Section 4.2 to show that the discontinuity of CROWN and hBox manifest already for small networks.

# C  Omitted Details on Continuity

## C.1  Network Used to Show Discontinuities

The sketch of the network described in Section 4.2, used to show discontinuities of hBox and CROWN relaxations is given in Figure 8.

## C.2  Proof for Continuity of IBP and DeepZ

We expand on the proof sketch given in the main paper to provide a complete proof of Theorem 2.

*Proof.* **IBP**: Recall that for IBP, $\boldsymbol{l}_i$ and $\boldsymbol{u}_i$ are computed directly as a function of $\boldsymbol{l}_{i-1}$, $\boldsymbol{u}_{i-1}$, and $\boldsymbol{\theta}$. For affine layers, this function is a sum of products of elements in $\boldsymbol{l}_{i-1}$, $\boldsymbol{u}_{-1}$ and $\boldsymbol{\theta}$, which is continuous w.r.t. all variables. If $i$ is a ReLU layer, the lower (upper) bound function is $\text{ReLU}(l_{i-1,j})$ (resp. $\text{ReLU}(u_{i-1,j})$), which is also clearly continuous. As compositions, sums, and products of continuous functions are continuous functions, this directly shows that $l_{L,j}$ are ultimately continuous.

**DeepZ**: For DeepZ, computing each $\boldsymbol{l}_i$ and $\boldsymbol{u}_i$ includes backsubstitution, where to obtain the final expressions (as in Equation 3), we repeatedly substitute in lower/upper bound expressions, based on the values of $\boldsymbol{\theta}$ and $\boldsymbol{l}_{i'}$ and $\boldsymbol{u}_{i'}$ from *all* previous layers. As before, it suffices to show that for some $i$, $\boldsymbol{l}_i$ and $\boldsymbol{u}_i$ are continuous w.r.t. $\boldsymbol{\theta}$ and all previous $\boldsymbol{l}_{i'}$ and $\boldsymbol{u}_{i'}$.

First, recall that during each step of backsubstitution we encounter terms of the form $\alpha \cdot x_{i',j}$, and based on the sign of $\alpha$ substitute the lower or the upper bound expression for $x_{i',j}$. When one such $\alpha = 0$, $\alpha \cdot x_{i',j}$ is continuous (both the left and the right limit equal the function value at that point, 0). Thus, we can reduce these cases to cases where no $\alpha$ values encountered are zero, i.e., all choices for the upper/lower bound to be substituted during backsubstitution are *fixed*.

Next, recall that even if we fix this choice, the actual expression we substitute in for the upper or lower bound may depend on the ReLU stability case. If all $\boldsymbol{l}_{i'}$ and $\boldsymbol{u}_{i'}$ are nonzero the stability is fixed, and by substituting affine and ReLU relaxation bounds during backsubstitution we can arrive at a closed form expression w.r.t. $\boldsymbol{\theta}$ that uses only elementary operations, which is continuous.

It is left to discuss the behavior in points where some elements of some $\boldsymbol{l}_{i'}$ or $\boldsymbol{u}_{i'}$ are zero. In this case the ReLU is still stable, but switches to being unstable on one side of zero. If $l_{i',j} = 0$ both upper and lower bound expressions for $x_{i'+1,j}$ are $x_{i',j}$, which is also the right limit. In the left limit, we use the unstable ReLU bounds $\lambda x_{i',j}$ and $\lambda x_{i',j} - \lambda l_{i',j}$ and see that for $l_{i',j} \to 0$, $\lambda \to 1$, and thus these bounds approach $x_{i',j}$ as well, so there is no discontinuity. Similarly, for $u_{i',j} = 0$ (the other stable case) the bounds (as well as the left limit) are 0. For the right limit, we again have the unstable case, but now $\lambda \to 0$, so both bounds approach 0, implying that this is also not a discontinuity.

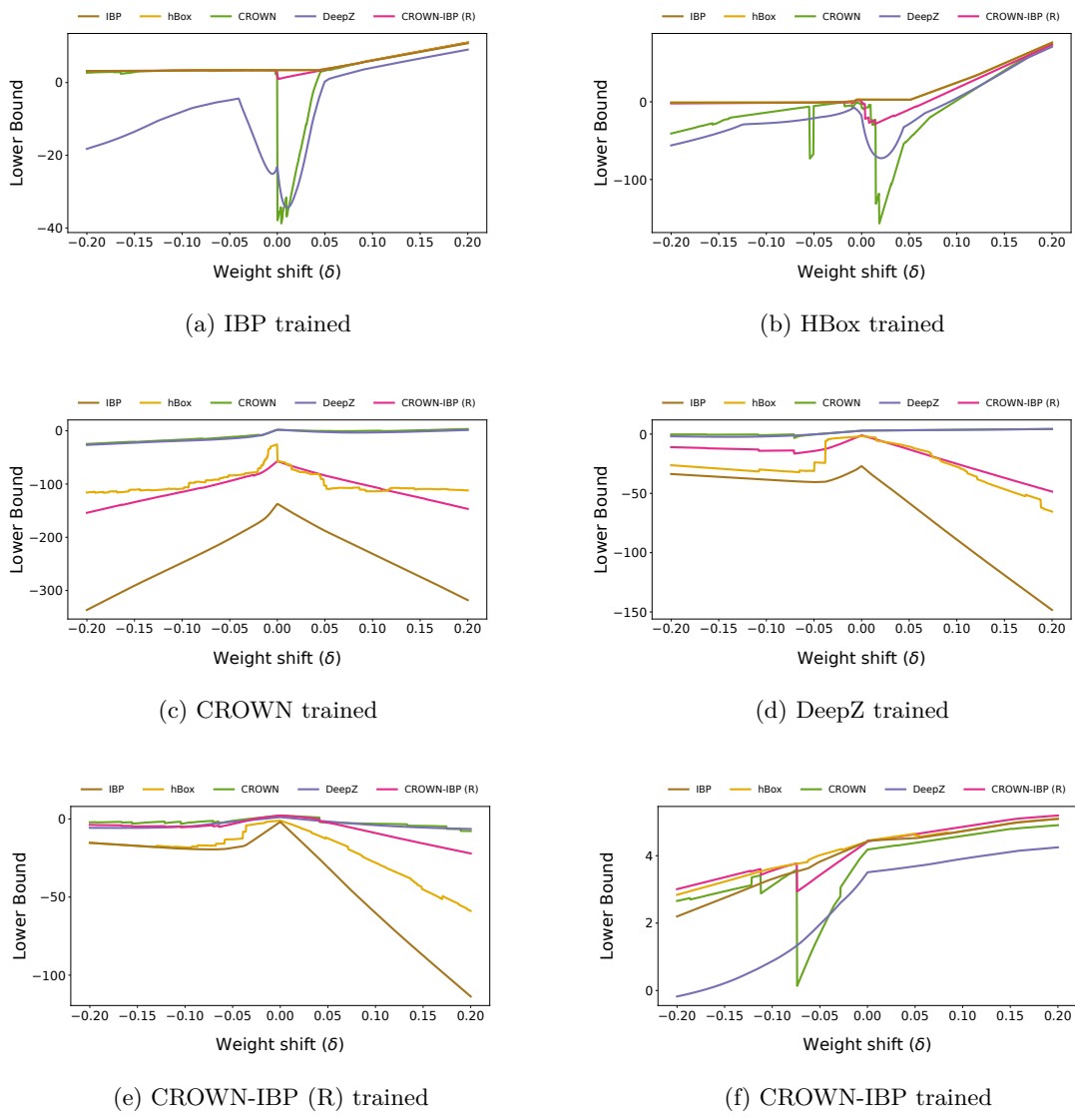

(a) IBP trained

(b) HBox trained

(c) CROWN trained

(d) DeepZ trained

(e) CROWN-IBP (R) trained

(f) CROWN-IBP trained

Figure 9: Lower bounds of convex relaxations for 5 FC networks from Table 2 with $\epsilon = \epsilon_{test} = 0.3$, and a network trained with CROWN-IBP, showing continuity and sensitivity of each relaxation. Each subfigure shows the name of the relaxation used to train the corresponding network.

To conclude, we showed that $\boldsymbol{l}_i$ and $\boldsymbol{u}_i$ are continuous w.r.t. $\boldsymbol{\theta}$ and all previous $\boldsymbol{l}_{i'}$ and $\boldsymbol{u}_{i'}$. This can be composed to conclude that $l_{L,j}$ is continuous w.r.t. $\boldsymbol{\theta}$, using the same argument we used for IBP. $\qquad \square$

# D   Details of Gradient Descent Experiments

In this section we provide details of the experiments with used to generate Figure 4, discussed in Section 4.4. For both examples we consider an architecture consisting of 2 hidden layers with 10 neurons each, and 2 output neurons. The network receives a 1-dimensional input. We randomly sample the input, weights, and biases as integers in $[-4, 4]$, and sample the perturbation radius $\epsilon$ between 0.1 and 4.1. For the experiment with the discontinuous CROWN relaxation, we set the initial learning rate to 0.02, learning rate decay to 0.99, and ran for 20 epochs. For the experiment with the sensitive DeepZ relaxation, we set the initial learning rate to 0.005, learning rate decay to 0.99, and ran for 100 epochs. We sampled a number of different networks for both scenarios, and chose the one which best illustrates the behavior of gradient descent.

## E    Continuity and Sensitivity in Practice

We present more results and discuss the details of experiments presented in Figure 5 and Figure 6.

### E.1    Additional Figures

Here we show additional figures for continuity and sensitivity with same setup as for Figure 5. In Figure 9 we show the bounds for each of the MNIST FC networks from Table 2, using $\epsilon = \epsilon_{test} = 0.3$, and a network trained with CROWN-IBP in the same setup. All plots are generated on the same example. We can highlight some differences compared to the naturally trained network. First, as explained earlier, the relaxation used for training typically obtains the tightest bounds. Next, we can see that if the network was trained with CROWN or hBox, there is a significantly smaller number of discontinuities than in the cases when the network was trained naturally or using some other relaxation. Even though there is a lack of discontinuities in these cases, these networks do not perform well (see Table 2) which suggests that, while the network learned to eliminate the discontinuities, the performance was still hurt by them earlier in the training. Finally, we see that evaluating IBP and hBox trained models using the DeepZ relaxation shows its increased sensitivity.

### E.2    Measuring Continuity and Sensitivity in Training

We provide details on the experiment discussed in Section 5.1 where we measure continuity and sensitivity of relaxations during certified training. For this experiment, we use the FC network and train on MNIST, using the same hyperparameters used to produce our main results. We measure sensitivity at every training step for the first 4 epochs, and once per epoch afterwards. Continuity is measured every 50 steps. Next, we describe the exact methods used to compute the two quantities shown in Figure 6.

To measure continuity, we compute the gradient $\nabla_\theta \mathcal{L}$ w.r.t. current parameters $\theta$, where $\mathcal{L}$ is the loss that each relaxation attempts to minimize at the current epoch for a fixed input sample. Then, we consider the line segment between $\theta - \alpha \nabla_\theta \mathcal{L}$ and $\theta + \alpha \nabla_\theta \mathcal{L}$ for $\alpha = 0.03$. We discretize this line segment into $n = 500$ parameter points $\theta_1, \theta_2, ..., \theta_n$. For each parameter value, we compute the loss values $l_1, l_2, ..., l_n$, and define the differences $\Delta_i = |l_{i+1} - l_i|$. Intuitively, there is a discontinuity between $\theta_i$ and $\theta_{i+1}$ if $\Delta_i$ is significantly bigger than its neighbors $\Delta_{i-1}$ and $\Delta_{i+1}$. Formally, we say there is a discontinuity if there exists $i \in \{1, \ldots, n-2\}$ for which we have that $\Delta_i > C \cdot \Delta_{i-1} + D$ and $\Delta_i > C \cdot \Delta_{i+1} + D$, where we set $C = 10$ and $D = 10^{-5}$. We measure discontinuity using this approach for a batch of 100 input samples and in Figure 6 report the proportion of samples inside the batch for which we have found a discontinuity. For IBP and DeepZ, we could also use our theoretical results from Section 4.2 proving that they are always continuous.

When computing sensitivity, recall from Section 4.3 that IBP and hBox always have the trivial sensitivity of 1 which corresponds to log sensitivity of 0. For CROWN-IBP (R), we previously derived worst case sensitivity of $\mathcal{O}(M^{B+1})$, where $B$ is the number of ReLU-affine blocks in the network. This bound assumes that all ReLU layers are unstable (meaning they contain at least one unstable ReLU neuron), which usually holds for trained networks and practical values of $\epsilon_{test}$. As this is not always the case when observing the whole training procedure, we extend the previous analysis with the observation that a sequence of consecutive affine and stable ReLU layers can be treated as a single affine layer, and obtain a tighter sensitivity upper bound of $\mathcal{O}(M^{B'+1})$, where $B'$ denotes the number of ReLU-affine blocks where the ReLU layer is unstable. In Figure 6 we report the log sensitivity $B' + 1$, averaged across all samples in a single batch. Similarly, for DeepZ and CROWN, their sensitivity is $\mathcal{O}(3^{B'} M^{B'+1})$, taking logarithm and factoring out $\log M$, we obtain their log sensitivity is $(B' + 1) \cdot (1 + \frac{\log 3}{\log M})$. We set $M = 400$ as this is the biggest number of neurons in a layer for the network in this experiment.

## F    Details and Additional Results of Main Experiments

We provide all omitted details of our main experiments given in Section 5.2, including details of networks and training parameters (Appendix F.1), additional investigations into the effect of the seed (Appendix F.2), and certifying with different relaxations to those used in training (Appendix F.3), as well as complete results omitted from the main text (Appendix F.4), including the hybrid CROWN-IBP defense.

Table 7: The architectures used in our main experiments. "FC $n$" denotes a dense (fully-connected) layer with $n$ neurons. "CONV $k$ $w \times h + s$" denotes a convolutional layer with $k$ kernels of size $w \times h$ and stride $s$. All activations are ReLU. CONV+ is equivalent to „small" in Gowal et al. (2018).

| FC | CONV | CONV+ |
|---|---|---|
| FC 400 | CONV 16 4x4+2 | CONV 16 4x4+2 |
| FC 200 | FC 100 | CONV 32 4x4+1 |
| FC 100 | FC 10 | FC 100 |
| FC 100 | | FC 10 |
| FC 10 | | |

Table 8: The best choice of hyperparameters for each MNIST model in our evaluation.

| Net | $\epsilon_{test}$ | Method | $\epsilon_{train}$ | $N_w$ | $N_r$ | $\kappa_{end}$ | $\lambda$ | $\alpha$ | LR schedule | Elision |
|---|---|---|---|---|---|---|---|---|---|---|
| FC | 0.1 | IBP | 0.2 | 10 | 100 | 0.5 | 5e-6 | 5e-4 | milestones | yes |
| FC | 0.2 | IBP | 0.2 | 10 | 100 | 0 | 5e-6 | 5e-4 | milestones | yes |
| FC | 0.3 | IBP | 0.3 | 10 | 100 | 0 | 5e-6 | 5e-4 | milestones | yes |
| FC | 0.1 | hBox | 0.1 | 0 | 50 | 0 | 5e-5 | 5e-4 | milestones | yes |
| FC | 0.2 | hBox | 0.2 | 10 | 50 | 0.5 | 5e-5 | 5e-4 | milestones | yes |
| FC | 0.3 | hBox | 0.3 | 0 | 50 | 0.5 | 5e-5 | 5e-4 | milestones | yes |
| FC | 0.1 | CROWN | 0.1 | 0 | 100 | 0 | 0 | 5e-4 | milestones | yes |
| FC | 0.2 | CROWN | 0.2 | 10 | 50 | 0 | 0 | 5e-4 | milestones | yes |
| FC | 0.3 | CROWN | 0.3 | 10 | 100 | 0 | 0 | 5e-4 | milestones | yes |
| FC | 0.1 | DeepZ | 0.1 | 10 | 50 | 0 | 5e-6 | 5e-4 | milestones | yes |
| FC | 0.2 | DeepZ | 0.2 | 0 | 50 | 0 | 5e-6 | 1e-3 | steps | no |
| FC | 0.3 | DeepZ | 0.3 | 0 | 50 | 0 | 5e-6 | 1e-3 | steps | no |
| FC | 0.1 | CROWN-IBP (R) | 0.2 | 0 | 50 | 0.5 | 5e-6 | 5e-4 | milestones | yes |
| FC | 0.2 | CROWN-IBP (R) | 0.3 | 0 | 50 | 0.5 | 5e-6 | 5e-4 | milestones | yes |
| FC | 0.3 | CROWN-IBP (R) | 0.3 | 0 | 50 | 0.5 | 5e-6 | 5e-4 | milestones | yes |
| FC | 0.1 | CROWN-IBP | 0.2 | 10 | 50 | 0.5 | 0 | 5e-4 | milestones | yes |
| FC | 0.2 | CROWN-IBP | 0.2 | 10 | 50 | 0 | 0 | 5e-4 | milestones | yes |
| FC | 0.3 | CROWN-IBP | 0.3 | 10 | 100 | 0 | 0 | 5e-4 | milestones | yes |
| CONV | 0.1 | IBP | 0.2 | 0 | 50 | 0.5 | 5e-6 | 5e-4 | milestones | yes |
| CONV | 0.2 | IBP | 0.3 | 0 | 50 | 0.5 | 5e-6 | 5e-4 | milestones | yes |
| CONV | 0.3 | IBP | 0.3 | 0 | 50 | 0.5 | 5e-6 | 5e-4 | milestones | yes |
| CONV | 0.1 | hBox | 0.3 | 0 | 50 | 0.5 | 5e-5 | 5e-4 | milestones | yes |
| CONV | 0.2 | hBox | 0.3 | 0 | 50 | 0.5 | 5e-5 | 5e-4 | milestones | yes |
| CONV | 0.3 | hBox | 0.3 | 0 | 50 | 0.5 | 5e-5 | 5e-4 | milestones | yes |
| CONV | 0.1 | CROWN | 0.2 | 10 | 100 | 0 | 0 | 5e-4 | milestones | yes |
| CONV | 0.2 | CROWN | 0.2 | 10 | 100 | 0 | 0 | 5e-4 | milestones | yes |
| CONV | 0.3 | CROWN | 0.3 | 10 | 100 | 0 | 0 | 5e-4 | milestones | yes |
| CONV | 0.1 | DeepZ | 0.2 | 0 | 50 | 0 | 5e-6 | 5e-4 | milestones | yes |
| CONV | 0.2 | DeepZ | 0.2 | 0 | 50 | 0 | 5e-6 | 5e-4 | milestones | yes |
| CONV | 0.3 | DeepZ | 0.3 | 0 | 50 | 0 | 5e-5 | 5e-4 | milestones | yes |
| CONV | 0.1 | CROWN-IBP (R) | 0.1 | 10 | 50 | 0 | 0 | 5e-4 | milestones | yes |
| CONV | 0.2 | CROWN-IBP (R) | 0.2 | 10 | 50 | 0 | 0 | 5e-4 | milestones | yes |
| CONV | 0.3 | CROWN-IBP (R) | 0.3 | 10 | 50 | 0.5 | 5e-6 | 5e-4 | milestones | yes |
| CONV | 0.1 | CROWN-IBP | 0.2 | 10 | 100 | 0.5 | 5e-6 | 5e-4 | milestones | yes |
| CONV | 0.2 | CROWN-IBP | 0.3 | 10 | 50 | 0.5 | 5e-6 | 5e-4 | milestones | yes |
| CONV | 0.3 | CROWN-IBP | 0.3 | 10 | 50 | 0 | 5e-6 | 5e-4 | milestones | yes |

### F.1 Networks and Hyperparameters

Here we detail the setup of the main experiments shown in Table 2. All runs use a single GeForce RTX 2080 Ti GPU. The details of networks FC, CONV, and CONV+ are shown in Table 7. The hyperparameters vary by dataset.

For MNIST, we tune all hyperparameters thoroughly. We train all models for 200 epochs, starting with a *warm-up* ($N_w$ epochs) followed by a *ramp-up* period ($N_r$ epochs) to stabilize the training procedure (Gowal et al., 2018). During the warm-up we train the network naturally. During the ramp-up we gradually increase the perturbation radius $\epsilon$ from 0 to $\epsilon_{train}$, decrease $\kappa$ from $\kappa_{start} = 1$ to $\kappa_{end}$ (shifting from natural to certified training), and for CROWN-IBP gradually shift from CROWN-IBP (R) to IBP loss. We use a batch size of 100 (50 for memory intensive models) and train using the Adam optimizer with the initial learning rate $\alpha$. Finally, we use $L_1$ regularization with the strength hyperparameter $\lambda$. We tune $(N_w, N_r, \kappa_{end}, \lambda, \alpha)$, as well as the learning rate schedule (*milestones*, where we reduce the learning rate $10\times$ at epochs 130 and 190, or *steps*, where we halve it every 20 epochs), and the choice of last layer elision (where we elide the final layer $h_L$ of the network with the specifications $\boldsymbol{c}_{y'}$ as in Gowal et al. (2018)). For each perturbation radius $\epsilon_{test} \in \{0.1, 0.2, 0.3\}$, we train with $\epsilon_{train} \in \{0.1, 0.2, 0.3, 0.4\}$ and report the best result. In Table 8 we show the best choice of hyperparameters for each model used in our evaluation (see Appendix F.4 for full results).

For FashionMNIST, we reuse the best hyperparameter choice of the corresponding MNIST model.

For SVHN, we use the parameters given in prior work as a starting point, and introduce minimal changes. For IBP, CROWN-IBP (R), and CROWN-IBP, we start from the parameters given in Gowal et al. (2018): we train for 2200 epochs with batch size 50, warm-up for 10 epochs, ramp-up for 1100 epochs, use Adam with initial learning rate of $\alpha = $ 1e-3 (reduced $10\times$ at 60% and 90% of the training steps), and use $\epsilon_{train} = 1.1\epsilon_{test}$. We do not use random translations (as we notice these harm the results on large $\epsilon_{test}$), and we tune $\kappa_{end}$ (trying 0 and 0.5 for each method—0 performs better for all methods except CROWN-IBP (R)), introduce L1 regularization (improves the results only for IBP, with $\lambda = $ 5e-5), tune the initial learning rate (we end up using $\alpha = $ 5e-4 for IBP and CROWN-IBP). For hBox, DeepZ and CROWN, we use the parameters from Wong & Kolter (2018): batch size of 20, training for 100 epochs (training longer does not improve the results), using Adam with initial learning rate $\alpha = $ 1e-3 halved every 10 epochs, ramp-up w.r.t. $\epsilon$ of 50 epochs where we start from $\epsilon = 0.001$. We introduce ramp-up w.r.t. $\kappa$ with $\kappa_{end} = 0$. As before, we exclude the data transformations. For all three methods we use L1 regularization with $\lambda = $ 5e-6.

For CIFAR-10, we similarly use the parameters from prior work. For IBP, CROWN-IBP (R), and CROWN-IBP, we use the values from Zhang et al. (2020): 3200 epochs, 320 of warm-up and 1600 of ramp-up (using $\kappa_{end} = 0$ and $\epsilon_{train} = 1.1\epsilon_{test}$ for all methods, $\kappa_{start} = 1$ for IBP and $\kappa_{start} = 0$ for other methods), Adam with $\alpha = $ 5e-4 reduced $10\times$ at epochs 2600 and 3040, and random horizontal flips and crops as augmentation. We halve the batch size to 512 for all three methods. For DeepZ and hBox we use 50 random Cauchy projections (Wong et al., 2018) and the parameters based on Wong et al. (2018) but with extended training length and introduced ramp-up w.r.t. $\kappa$: we train with batch size 50 for 240 epochs, 80 of which are ramp-up, using Adam optimizer with $\alpha = $ 5e-4, halved every 10 epochs. During ramp-up we start from $\epsilon = 0.001$, and use $\kappa_{start} = 1$ and $\kappa_{end} = 0$.

### F.2 Estimating the Effect of the Seed

To estimate variability and demonstrate that it does not significantly impact our main conclusions, we use one efficient method (IBP) and perform the same run with best parameters from Appendix F.1 with 10 seed values, across two datasets (MNIST and FashionMNIST) and both networks (FC and CONV). In all experiments we use $\epsilon_{test} = 0.3$. The results, with the mean and the standard deviation of obtained results, are given in Table 9. Note that, for both MNIST networks, the results we report in Table 2 are the best out of all 10 seeds (74% and 86.8% respectively), as expected given that the hyperparameters were tuned on this seed. As is it too expensive to run repetitions for relaxations other than IBP, we can not estimate the confidence intervals of their results. Nonetheless, if we took the confidence interval of the size of two standard deviations for our IBP results, it would not change the conclusions we made based on single experiment runs reported in Table 2. Namely, continuity and sensitivity, alongside tightness, can explain the results in Table 2.

Table 9: The variability of the results when changing the seed.

| Dataset | Net | Seed | | | | | | | | | | Mean | Stddev |
|---|---|---|---|---|---|---|---|---|---|---|---|---|---|
| | | 10 | 11 | 12 | 13 | 14 | 15 | 16 | 17 | 18 | 19 | | |
| MNIST | FC | 74 | 70.8 | 70.6 | 71.5 | 70.9 | 69.9 | 72.7 | 72.2 | 65 | 69 | 70.7 | 2.45 |
| MNIST | CONV | 86.8 | 86.6 | 85.7 | 86.5 | 86.1 | 86.2 | 86.1 | 85.4 | 86.3 | 85 | 86.1 | 0.56 |
| FashionMNIST | FC | 40.4 | 40.4 | 38.8 | 40.6 | 42.1 | 39.8 | 45 | 42.5 | 42.6 | 42.7 | 41.5 | 1.82 |
| FashionMNIST | CONV | 52 | 51.7 | 51 | 50 | 52.1 | 51.6 | 50.2 | 50.4 | 52.1 | 52.2 | 51.3 | 0.86 |

Table 10: The evaluation of models trained with certified training using different convex relaxations.

| Net | $\epsilon_{test}$ | Method (training) | Method (certification) | | | | |
|---|---|---|---|---|---|---|---|
| | | | IBP | hBox | CROWN | DeepZ | CROWN-IBP (R) |
| FC | 0.1 | IBP | 89.5 | 89.5 | 44.7 | 67 | 84.1 |
| FC | 0.3 | IBP | 74 | 74 | 3.7 | 25.5 | 64.5 |
| FC | 0.1 | hBox | 88.4 | 88.4 | 36.6 | 55.9 | 83.4 |
| FC | 0.3 | hBox | 57 | 57 | 2.4 | 7.1 | 10.8 |
| FC | 0.1 | CROWN | 0 | 0 | 90.9 | 91.6 | 0 |
| FC | 0.3 | CROWN | 0 | 0 | 54.3 | 57.3 | 0 |
| FC | 0.1 | DeepZ | 0 | 0.1 | 92.5 | **93** | 0.6 |
| FC | 0.3 | DeepZ | 0 | 0.1 | 64.2 | 63.5 | 1.8 |
| FC | 0.1 | CROWN-IBP (R) | 5.7 | 76.8 | 90.2 | 90.5 | 90.6 |
| FC | 0.3 | CROWN-IBP (R) | 1.8 | 10.4 | 37.4 | 56.8 | 70.5 |
| FC | 0.1 | CROWN-IBP | 91.2 | 91.2 | 65.3 | 78.2 | 88.2 |
| FC | 0.3 | CROWN-IBP | 77.9 | 77.9 | 8.6 | 29 | 72.6 |
| CONV | 0.1 | IBP | 94.6 | 94.7 | 92.3 | 92.9 | 93.1 |
| CONV | 0.3 | IBP | 86.8 | 86.8 | 12.4 | 46.7 | 64.9 |
| CONV | 0.1 | hBox | 92.7 | 92.7 | 90.6 | 91.2 | 91.7 |
| CONV | 0.3 | hBox | 83.7 | 83.7 | 26.3 | 50.3 | 73.4 |
| CONV | 0.1 | CROWN | 0.1 | 87.8 | 94.4 | 94.5 | 93.6 |
| CONV | 0.3 | CROWN | 0 | 0.3 | 65.2 | 70.2 | 55.1 |
| CONV | 0.1 | DeepZ | 12 | 92.6 | 94.9 | **95.1** | 94.7 |
| CONV | 0.3 | DeepZ | 0.4 | 25.8 | 69.8 | **74** | 60.9 |
| CONV | 0.1 | CROWN-IBP (R) | 0 | 59.1 | 93.1 | 93.5 | 93.4 |
| CONV | 0.3 | CROWN-IBP (R) | 0 | 0.2 | 59.5 | 67.6 | 75.4 |
| CONV | 0.1 | CROWN-IBP | 94.4 | 94.5 | 92.4 | 92.9 | 93.3 |
| CONV | 0.3 | CROWN-IBP | 86.6 | 86.6 | 32.8 | 60.3 | 78 |

Table 11: Complete evaluation results on the MNIST dataset.

| | Method | $\epsilon_{test} = 0.1$ | | | $\epsilon_{test} = 0.2$ | | | $\epsilon_{test} = 0.3$ | | |
|---|---|---|---|---|---|---|---|---|---|---|
| | | Acc (%) | PGD (%) | CR (%) | Acc (%) | PGD (%) | CR (%) | Acc (%) | PGD (%) | CR (%) |
| FC | IBP | 94.8 | 90.7 | 89.5 | 92.6 | 86.6 | 82.4 | 88.7 | 80.8 | 74.0 |
| | hBox | 95.6 | 90.6 | 88.4 | 93.2 | 82.1 | 76.6 | 89.2 | 65.7 | 57.0 |
| | CROWN | 98.6 | 94.6 | 91.6 | 93.0 | 85.5 | 80.6 | 84.9 | 71.7 | 57.3 |
| | DeepZ | 98.3 | 95.1 | 92.5 | 95.0 | 90.9 | 85.1 | 87.4 | 77.9 | 64.2 |
| | CROWN-IBP (R) | 94.9 | 91.8 | 90.6 | 92.2 | 84.6 | 80.9 | 92.2 | 79.9 | 70.5 |
| | CROWN-IBP | 95.5 | 92.0 | 91.2 | 93.3 | 88.6 | 86.0 | 90.8 | 83.6 | 77.9 |
| CONV | IBP | 97.2 | 95.0 | 94.6 | 95.9 | 92.3 | 91.3 | 95.9 | 89.7 | 86.8 |
| | hBox | 95.1 | 93.0 | 92.7 | 95.1 | 90.9 | 89.5 | 95.1 | 87.9 | 83.7 |
| | CROWN | 96.8 | 95.1 | 94.5 | 96.8 | 92.6 | 88.0 | 92.6 | 84.3 | 70.2 |
| | DeepZ | 97.0 | 95.7 | 94.9 | 97.0 | 94.0 | 88.8 | 92.5 | 87.0 | 69.8 |
| | CROWN-IBP (R) | 98.5 | 95.2 | 93.4 | 95.5 | 90.9 | 86.9 | 93.4 | 84.5 | 75.4 |
| | CROWN-IBP | 96.9 | 94.8 | 94.4 | 95.6 | 92.0 | 90.9 | 94.8 | 89.1 | 86.6 |

## F.3  Training and Certifying with Different Relaxations

In this section, we investigate the effect of varying the convex relaxation used to certify a network trained using some method $\mathcal{M}$, to justify our choice of using the same relaxation for training and certification in our main experiments.

We use the MNIST models with $\epsilon_{test} = 0.3$ from Table 2 and the corresponding models for $\epsilon_{test} = 0.1$ from the full results given in Appendix F.4, on both FC and CONV architectures. We evaluate their certified robustness using all five introduced methods. The results are given in Table 10. Observe that almost all IBP trained models have extremely low certified robustness when certified with DeepZ, even though it is tighter, and vice versa. This confirms our previous statement that training with $\mathcal{M}$ produces a network particularly suitable to certification with $\mathcal{M}$, and justifies our decision to focus on this case. The few exceptions, i.e., the instances where a method different than $\mathcal{M}$ achieved a better result (by more than the minimal 0.1% after rounding), are marked in bold. We can see that certification with tighter CROWN often slightly improves DeepZ-trained networks. However, this improvement mostly leaves the relative order of methods unchanged and does not affect our conclusions. Note that if we are interested in the highest certified robustness of an already trained model, the best approach is to always use more expensive certifiers (Tjeng et al., 2019; Singh et al., 2019a; Tjandraatmadja et al., 2020) which are not fast enough to be used in training. However, here we focus on analyzing training properties of a single relaxation, and not on maximizing certified robustness.

## F.4  Complete Results of Main Experiments

In Tables 11 to 13 we present complete certified training evaluation results, expanding the ones given in Section 5.2. In all tables, Acc denotes accuracy, PGD denotes empirical robustness against PGD attacks (we use 100 steps with step size 0.01), and CR denotes certified robustness. For MNIST/FashionMNIST datasets we include two smaller perturbation radii, $\epsilon_{test} = 0.1$ and $\epsilon_{test} = 0.2$. Note that the paradox of certified training can rarely be observed for such small radii, and we thus in the main paper focus on the challenging case of strong adversaries, i.e., $\epsilon_{test} = 0.3$ for MNIST/FashionMNIST and $\epsilon_{test} = 8/255$ for SVHN/CIFAR-10, as it most clearly illustrates the differences between relaxations. Further, this case is of greater interest as it is a well-established benchmark for robustness even outside of the area of *certified* robustness (Madry et al., 2018; Croce et al., 2020). To explain the unusually high *standard accuracy* of CROWN-IBP (R) in our CIFAR-10 experiments, note that it is the only method that performs better with $\kappa_{end} = 0.5$ (as opposed to $\kappa_{end} = 0$). All other methods could reach similar standard accuracy with $\kappa_{end} = 0.5$, but their certified robustness would drop.

Table 12: Complete evaluation results on the FashionMNIST dataset.

| | Method | $\epsilon_{test} = 0.1$ | | | $\epsilon_{test} = 0.2$ | | | $\epsilon_{test} = 0.3$ | | |
| | | Acc (%) | PGD (%) | CR (%) | Acc (%) | PGD (%) | CR (%) | Acc (%) | PGD (%) | CR (%) |
|---|---|---|---|---|---|---|---|---|---|---|
| FC | IBP | 79.2 | 71.4 | 67.9 | 73.7 | 65.2 | 57.9 | 54.8 | 46.6 | 40.4 |
| | hBox | 79.2 | 73.1 | 70.0 | 76.3 | 62.5 | 56.4 | 68.7 | 49.2 | 39.6 |
| | CROWN | 80.8 | 70.6 | 67.8 | 70.9 | 53.7 | 49.5 | 52.4 | 35.9 | 30.2 |
| | DeepZ | 81.2 | 73.7 | 70.2 | 71.2 | 57.6 | 51.6 | 51.0 | 39.2 | 35.0 |
| | CROWN-IBP (R) | 76.6 | 68.1 | 66.5 | 69.6 | 54.7 | 51.3 | 69.6 | 46.7 | 41.1 |
| | CROWN-IBP | 78.1 | 72.1 | 69.6 | 74.7 | 64.7 | 58.9 | 66.7 | 55.9 | 47.9 |
| CONV | IBP | 80.3 | 74.1 | 72.7 | 76.5 | 65.5 | 61.4 | 76.5 | 58.8 | 52.0 |
| | hBox | 72.9 | 67.0 | 65.2 | 72.9 | 61.3 | 57.4 | 72.9 | 53.6 | 47.1 |
| | CROWN | 71.9 | 64.3 | 62.8 | 71.9 | 55.2 | 49.4 | 56.4 | 39.9 | 31.5 |
| | DeepZ | 72.3 | 66.6 | 64.8 | 72.3 | 60.1 | 53.4 | 56.3 | 40.7 | 34.0 |
| | CROWN-IBP (R) | 81.0 | 72.5 | 69.9 | 71.8 | 58.2 | 54.5 | 68.9 | 50.2 | 40.0 |
| | CROWN-IBP | 80.0 | 73.5 | 72.1 | 74.7 | 63.7 | 60.2 | 65.3 | 56.5 | 50.9 |

Table 13: Complete evaluation results on SVHN dataset with the CONV network and $\epsilon_{test} = 8/255$ (left), and on CIFAR-10 dataset with the CONV+ network and $\epsilon_{test} = 8/255$ (right).

| Method | Acc (%) | PGD (%) | CR (%) |
|---|---|---|---|
| IBP | 41.6 | 30.9 | 28.9 |
| hBox | 35.0 | 26.9 | 23.6 |
| CROWN | 44.0 | 27.3 | 23.4 |
| DeepZ | 38.7 | 27.7 | 24.5 |
| CROWN-IBP (R) | 65.8 | 37.2 | 27.5 |
| CROWN-IBP | 43.9 | 32.6 | 29.3 |

| Method | Acc (%) | PGD (%) | CR (%) |
|---|---|---|---|
| IBP | 39.8 | 32.4 | 29.0 |
| hBox | 36.2 | 25.9 | 20.0 |
| CROWN | OOM | OOM | OOM |
| DeepZ | 36.5 | 28.6 | 22.8 |
| CROWN-IBP (R) | 37.4 | 28.7 | 24.3 |
| CROWN-IBP | 41.1 | 32.8 | 30.3 |

Table 14: The extension of our main experimental results in various alternative settings. All experiments are done on the MNIST dataset.

| | Method | Original | Kaiming | Ortho | Xavier | IBPInit | L2 | BigLR | TinyLR | SGD | 10% | 30% | 50% |
|---|---|---|---|---|---|---|---|---|---|---|---|---|---|
| FC | IBP | 74.0 | 61.7 | 71.2 | 67.3 | 67.6 | 71.0 | 61.1 | 74.3 | 23.8 | 49.0 | 64.3 | 69.4 |
| | hBox | 57.0 | 37.4 | 48.7 | 41.5 | 11.3 | 47.7 | 44.3 | 52.4 | 10.0 | 17.9 | 35.2 | 45.8 |
| | CROWN | 57.3 | 58.3 | 59.9 | 61.8 | 55.2 | 58.3 | 61.0 | 55.8 | 11.3 | 31.3 | 48.3 | 52.9 |
| | DeepZ | 64.2 | 60.1 | 63.4 | 63.1 | 64.1 | 64.2 | 60.9 | 59.1 | 11.3 | 42.1 | 60.7 | 64.0 |
| | CROWN-IBP (R) | 70.5 | 61.5 | 67.5 | 64.8 | 65.8 | 69.6 | 70.3 | 63.4 | 4.5 | 44.2 | 62.8 | 68.6 |
| | CROWN-IBP | 77.9 | 76.4 | 76.7 | 77.2 | 77.7 | 77.4 | 76.0 | 77.9 | 19.0 | 67.7 | 73.5 | 75.4 |
| CONV | IBP | 86.8 | 86.6 | 85.7 | 86.1 | 85.8 | 85.8 | 85.7 | 86.6 | 83.3 | 80.7 | 84.2 | 85.0 |
| | hBox | 83.7 | 82.4 | 69.3 | 69.5 | 83.1 | 85.0 | 69.0 | 85.9 | 80.7 | 81.8 | 83.4 | 85.3 |
| | CROWN | 70.2 | 70.1 | 70.5 | 67.1 | 68.6 | 70.6 | 67.5 | 68.1 | 67.8 | 61.6 | 68.5 | 67.3 |
| | DeepZ | 69.8 | 69.7 | 67.1 | 67.6 | 69.0 | 69.5 | 66.7 | 67.8 | 67.2 | 62.6 | 67.9 | 68.5 |
| | CROWN-IBP (R) | 75.4 | 77.9 | 76.7 | 77.7 | 73.7 | 73.9 | 72.4 | 73.5 | 72.0 | 58.5 | 69.2 | 71.9 |
| | CROWN-IBP | 86.6 | 87.0 | 86.8 | 86.7 | 86.5 | 86.4 | 86.5 | 86.3 | 82.8 | 78.4 | 84.7 | 85.5 |

## G  Excluding Alternative Explanations

Here we extend a subset of our evaluation given in Section 5.2 to a wider range of settings, aiming to show that our claims about continuity and sensitivity of existing relaxations are not strictly dependent on one of the concrete settings used in our main experiments. We use MNIST and both FC and CONV networks.

The results are presented in Table 14. Each column represents evaluation with one modified setting compared to the base run from Section 5.2, repeated in the first column. The next four columns correspond to different weight initializations: Kaiming Normal (He et al.), Orthogonal (Saxe et al., 2014), Xavier Normal (Glorot & Bengio, 2010), and IBPInit (Shi et al., 2021). The column marked L2 indicates the use of L2 regularization instead of L1. For the following two columns, *BigLR* and *TinyLR*, we increase (respectively, decrease), the learning rate two times. We further experiment with using the SGD optimizer instead of Adam, and training on stratified subsets of the training set with 10, 30, and 50 percent of data, respectively. We can see that the relative order of relaxations is fairly consistent across all settings, implying that our conclusions from Section 5.2 hold and are not tied to a specific choice of a setting.

## H  Details of Relaxation Modifications

We give more details on modifications of relaxations explored in Section 6. For each, we describe the relaxation used as a base (also shown in the diagram in Figure 10), the performed change, the intended effect on properties, and the resulting unintended effect on properties. The summary of CR-AUC scores for all modifications is given in Table 15, produced by an experiment in the same setting as Figure 2. We treat the relaxation as loose if its tightness is closer to IBP than hBox. While we choose names indicative of the origin of each modification, note that we rediscover some less prominent relaxations considered in prior work, as an example, DeepZ-Diag corresponds to zDiag from (Mirman et al., 2018). In the following, when describing linear bounds, we use $x', x, l, u$ to refer to $x_{i,j}, x_{i-1,j}, l_{i-1,j}, u_{i-1,j}$ for brevity. Further, we refer to two stable ReLU cases as *positive* and *negative*.

**hBox-based Modifications.** Starting from hBox ▣ , and setting $x \leq x' \leq x - l$ (unstable case) produces a theoretically non-comparable *hBox-Diag* ▣ which is empirically much less tight. Going further to remove the discontinuity by setting the same bounds for the negative case, we arrive at *hBox-Diag-C* ▣ , completely sacrificing tightness (0.00 AUC in Table 15). Finally, we partially remedy tightness by switching between hBox and hBox-Diag using a heuristic similar to one in CROWN to get *hBox-Switch* ▣ , which as a side-effect introduces a discontinuity.

**DeepZ-based Modifications.** Fixing $0 \leq x' \leq u$ and $x \leq x' \leq x - l$ for the unstable case in DeepZ ▣ produces *DeepZ-Box* ▣ and *DeepZ-Diag* ▣ , both now not sensitive, but looser and discontinuous. We construct *DeepZ-Diag-C* ▣ and *DeepZ-Switch* ▣ analogously to hBox-Diag-C and hBox-Switch. Further, we introduce *DeepZ-Soft* ▣ to eliminate the discontinuity of DeepZ-Switch, by setting $\lambda = \sigma(\gamma(l/u - u/l))$ with parameter $\gamma$, where $\sigma$ denotes the sigmoid function. Then, for the unstable case we set $\lambda x \leq x' \leq \lambda x - \lambda l$ when $\lambda \geq u/(u - l)$ and $\lambda x \leq x' \leq \lambda x + u - \lambda u$ otherwise. We find that $\gamma = 1$ works best. While this both resolves the discontinuity and makes DeepZ-Switch tighter, it reintroduces the problem of sensitivity, leading to properties comparable to original DeepZ. Finally, we explore *DeepZ-IBP (R)* ▣ , a relaxation analogous to CROWN-IBP (R) using IBP for intermediate bounds, which partially alleviates the sensitivity issue of DeepZ by slightly sacrificing tightness.

**CROWN-based Modifications.** We obtain *CROWN-0* ▣ and *CROWN-1* ▣ by fixing $0 \leq x'$ (and $x \leq x'$ respectively) as the lower bound for the unstable case, in an attempt to remove the discontinuity caused by heuristically switching between them in CROWN ▣ . However, this introduces a new kind of discontinuities at $l = 0$ and $u = 0$ respectively and slightly harms tightness, making these relaxations still both discontinuous and highly sensitive. If we aim to obtain continuity, we can set $l \leq x'$ for the positive case in CROWN-0 to obtain *CROWN-0-C* ▣ , and analogously set $x \leq x'$ for the negative case in CROWN-1 to get *CROWN-1-C* ▣ . While the latter becomes prohibitively loose, the former retains some tightness (i.e., it is tighter than IBP). Alternatively, modifying CROWN-0 and CROWN-1 to alleviate high sensitivity instead of discontinuity by setting $x' \leq x - l$ for the unstable case of CROWN-0, or $x' \leq u$ for the unstable case of CROWN-1, we obtain *CROWN-0-Tria* ▣ and *CROWN-1-Tria* ▣ , both looser than their original

Table 15: Tightness of relaxation modifications quantified as CR-AUC, calculated in the setting of Figure 2.

| | IBP | hBox | hBox-Diag | hBox-Diag-C | hBox-Switch | DeepZ | DeepZ-Box | DeepZ-Diag | DeepZ-Diag-C | DeepZ-Switch | DeepZ-Soft | DeepZ-IBP (R) | CROWN | CROWN-0 | CROWN-0-C | CROWN-0-Tria | CROWN-0-Tria-C | CROWN-1 | CROWN-1-C | CROWN-1-Tria | CROWN-1-Tria-C | CROWN-Soft | CROWN-IBP (R) | CROWN-Soft-IBP |
|---|---|---|---|---|---|---|---|---|---|---|---|---|---|---|---|---|---|---|---|---|---|---|---|---|
| CR-AUC | 0.73 | 1.76 | 1.11 | 0.00 | 1.96 | 3.00 | 2.07 | 1.38 | 0.44 | 2.37 | 2.75 | 1.98 | 3.36 | 3.01 | 1.31 | 1.59 | 0.22 | 2.11 | 0.00 | 1.67 | 0.00 | 3.24 | 2.15 | 2.15 |

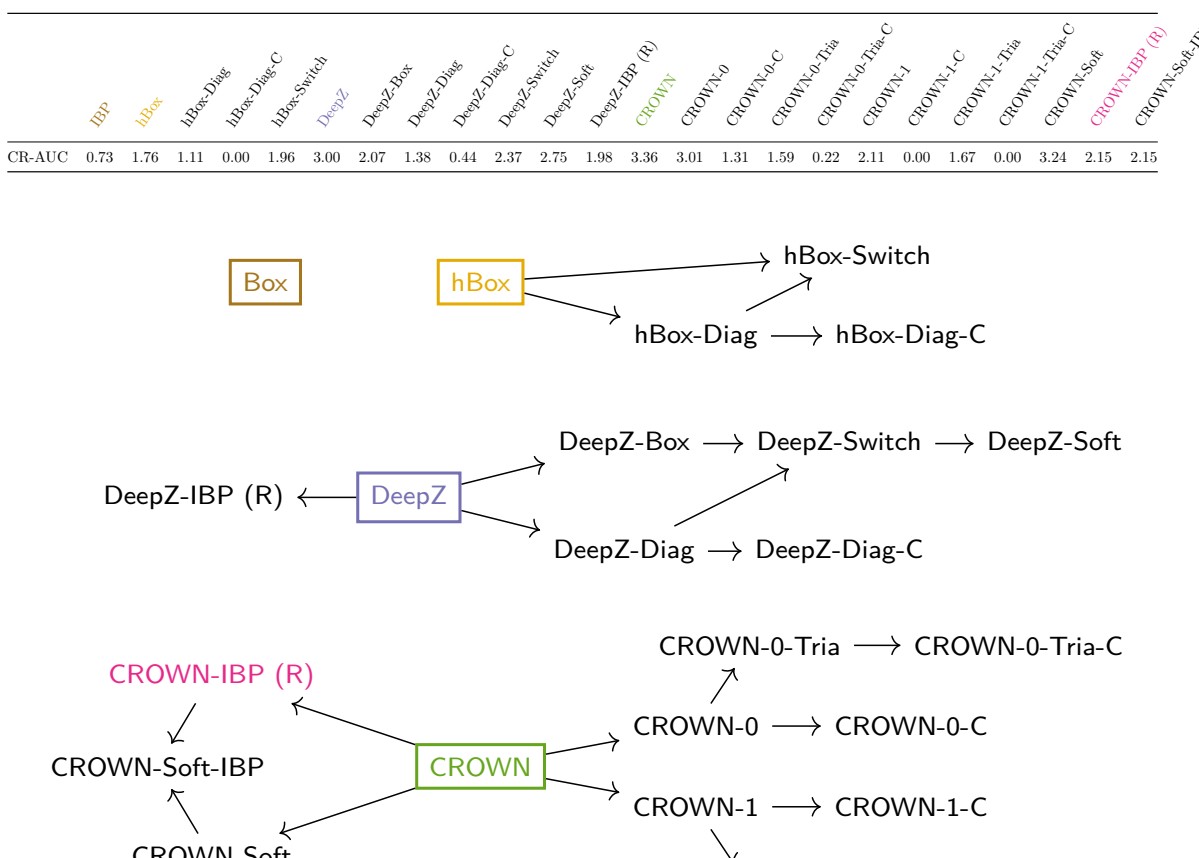

Figure 10: A display of explored relaxation modifications indicating the origin of each proposed modification.

relaxations, as can be seen in Table 15. These two relaxations can be further made continuous: using $x' \leq x - l$ for the negative and $l \leq x'$ for the positive case in CROWN-0-Tria leads to *CROWN-0-Tria-C* , and using $x \leq x'$ for the negative and $x' \leq u$ for the positive case in CROWN-1-Tria leads to *CROWN-1-Tria-C* , both sacrificing most of their tightness to obtain two other favorable properties. Further, we investigate the continuous variant of CROWN, *CROWN-Soft* , obtained analogously to DeepZ-Soft, and *CROWN-Soft-IBP* obtained by combining both the ideas of soft switching from CROWN-Soft and computing intermediate bounds with IBP in CROWN-IBP (R) .

## I   Generalizing Continuity and Sensitivity

We now discuss how continuity and sensitivity could be generalized from binary to numerical metrics to compare any pair of relaxations. For continuity, this can be done by empirically measuring the number of discontinuities along one direction in the loss landscape as we have done in Figure 5 and Figure 6 (left). These experiments show that CROWN indeed has more discontinuities than CROWN-IBP (R). Similarly, sensitivity can be generalized by considering the exact upper bound degree (instead of only whether it is greater than 1), as derived in Section 4.3. This would explain that DeepZ, which has the sensitivity of $O(3^B M^{B+1})$, performs worse than CROWN-IBP (R) which has the sensitivity of $O(M^{B+1})$. Overall, while our properties were designed to compare IBP with other relaxations, they could be extended to compare any pair of relaxations.

## J   Detailed Derivations of Sensitivity

In this section we provide more detailed derivations of sensitivity, expanding on those presented in Section 4.3.

As the first layer is always affine, we have $x_{1,j} \in P_1(\delta)$ for all relaxations. To compute the sensitivity of each relaxation, we will sequentially analyze the effect of each network layer.

**IBP/hBox.** For IBP, assume that at layer $i$, all $x_{i-1,j} \in P_N(\delta)$. For an affine layer, as $l_{i,j}$ and $u_{i,j}$ are linear combinations of elements of $\boldsymbol{u}_{i-1}$ and $\boldsymbol{l}_{i-1}$, the degree stays the same and $x_{i,j} \in P_N(\delta)$. For a ReLU layer, as $u_{i,j} = \mathrm{ReLU}(u_{i-1,j})$ (same for $l_{i,j}$) we again have $x_{i,j} \in P_N(\delta)$ (we consider two cases: $u_{i,j} = 0$ and $u_{i,j} = u_{i-1,j}$, both of which are in $P_N(\delta)$). Thus, since the degree does not change neither in affine nor ReLU operations, and in the first layer the degree is 1, all neurons are in $P_1(\delta) \subseteq R_1(\delta)$, i.e., the sensitivity of IBP is 1. For hBox, the only difference are affine layers, where now $x_{i,j} = (\boldsymbol{W}_i \boldsymbol{x}_{i-1})_j + b_{i,j}$. As linear combinations of elements of $P_N(\delta)$ are in $P_N(\delta)$, all neurons stay in $P_1(\delta)$ and the sensitivity of hBox is also 1.

**DeepZ/CROWN.** The ReLU bounds of DeepZ, $\lambda x_{i-1,j}$ and $\lambda x_{i-1,j} - \lambda l_{i-1,j}$, significantly increase the sensitivity. Recall that here $\lambda := u_{i-1,j}/(u_{i-1,j} - l_{i-1,j})$. After the first ReLU layer, we have that $x_{2,j} \in R_2(\delta)$ because $\lambda \in R_1(\delta)$ and $x_{1,j} \in P_1(\delta)$. This changes the behavior of all following affine layers, as a linear combination of $M$ elements of $R_N(\delta)$ is in $R_{MN}(\delta)$. Thus, $x_{3,j} \in R_{2M}(\delta)$. For the following ReLU layers, if we assume the inputs $x_{i-1,j}$ are in $R_N(\delta)$, we have that $\lambda \in R_{2N}(\delta)$, and thus the outputs $x_{i,j}$ are in $R_{3N}(\delta)$. This is because we are multiplying elements of $R_N(\delta)$ and $R_{2N}(\delta)$, and we get an expression in $R_{3N}(\delta)$. Putting this together, each ReLU-affine block from layer 4 onwards multiplies the sensitivity by $3M$. As there are $B \equiv \lfloor L/2 \rfloor - 1$ such blocks, we obtain $2 \cdot 3^B M^{B+1}$ for the final sensitivity. Recall that CROWN has the same upper bound as DeepZ for unstable ReLUs, but chooses the lower bound adaptively: $0 \leq x_{i,j}$ if $-l_{i-1,j} \geq u_{i-1,j}$, or $x_{i-1,j} \leq x_{i,j}$ otherwise. Both of these options do not change the degree: if $x_{i-1,j}$ is in $R_N(\delta)$, then also $x_{i,j}$ is in $R_N(\delta)$. However, CROWN uses the same upper ReLU bounds as DeepZ with the slope $\lambda := u_{i-1,j}/(u_{i-1,j} - l_{i-1,j})$, so we can apply the same analysis as before for the upper bound, and show that the sensitivity of CROWN is $2 \cdot 3^B M^{B+1}$ as well, despite having a simpler expression for the lower bound. Thus, both DeepZ and CROWN are highly sensitive.

**CROWN-IBP (R).** Here, at ReLU layer $i$ during the (only) backsubstitution, we have to consider $l_{i-1,j}$ and $u_{i-1,j}$ separately from $x_{i-1,j}$. Recall that the former were precomputed with IBP, and are thus in $P_1(\delta)$ (using the same analysis explained earlier for IBP). Then, $x_{i-1,j}$ get substituted as usual for CROWN and can carry larger sensitivity. The main difference here is that $\lambda := u_{i-1,j}/(u_{i-1,j} - l_{i-1,j})$ is in $R_1(\delta)$ because $l_{i-1,j}$ and $u_{i-1,j}$ were computed using IBP, meaning they are in $P_1(\delta)$, and this implies that $\lambda \in R_1(\delta)$. Assuming $x_{i-1,j} \in R_N(\delta)$ $(N \geq 1)$ and using the previous observation that we always have $\lambda \in R_1(\delta)$, it follows that $x_{i,j} \in R_{N+1}(\delta)$. As the affine layers have the same effect as in the case of CROWN, each of $B$ following ReLU-affine blocks now increases sensitivity from $N$ to $g(N) = (N+1)M$, and we have, as before, $x_{3,j} \in R_{2M}(\delta)$. Thus, the final sensitivity is $g^B(2M) = 2M^{B+1} + M^B + \ldots + M$, which can be proven by induction. We have $g^1(2M) = 2M^2 + M$ and assume the identity holds for $g^{B-1}(2M)$. Now:

$$g^B(2M) = (g^{B-1}(2M) + 1)M$$
$$= (2M^B + M^{B-1} + \ldots + M + 1)M$$
$$= (2M^{B+1} + M^B + \ldots + M),$$

completing the proof. We further write $g^B(2M) = M^{B+1} - 1 + (M^{B+1} + M^B + \ldots + M + 1)$ and using the closed-form formula for the sum of first $B + 2$ terms of a geometric series (assuming $M \neq 1$) simplify this to:

$$g^B(2M) = M^{B+1} - 1 + \frac{M^{B+2} - 1}{M - 1} = \frac{2M^{B+2} - M^{B+1} - M}{M - 1}, \tag{6}$$

which is in $\mathcal{O}(M^{B+1})$. Clearly, CROWN-IBP (R) is also significantly more sensitive than IBP and hBox.

