# OpenReview forum: "On the Paradox of Certified Training"
_TMLR — Accepted by TMLR_

### Review · Reviewer_BLjF · 2022-07-22

**Summary Of Contributions:**

The paper addresses a so-called paradox in certified robust training of neural networks, where tighter convex relaxations tend to produce lower certified robustness. This phenomenon is counterintuitive; tighter relaxations serve as better approximations to the network layers, and one would expect that training a network with such relaxations would produce better results.

The paper proposes a possible explanation for this effect. Tighter relaxations tend to produce undesirable properties which hinder training algorithms. Specifically, the authors identify two main properties:

- Discontinues: the training loss function, after the relaxation, exhibits discontinues. This is clearly unfavorable for gradient-based algorithms and will have a negative impact on the training algorithm.

- Sensitivity: The effect of small weight changes on the output loss. This affects the stability of the training algorithm, and stable algorithms tend to perform better.

The authors show, through numerical evidence, that tighter relaxations tend to either posses many discontinues of the loss function or have large sensitivity (or both). This provides a possible way to explain the apparent paradox. It is not the tightness of the relaxation that matters but rather its interplay with the training procedure.

**Broader Impact Concerns:**

I do not have concerns about the broader impact.

**Requested Changes:**

As I indicated above, the claims made in the paper are mostly supported by convincing and clear evidence, and I do think that some of TMLR's audience would be interested in the findings. So, conditional on addressing the issues I raised above, I would be inclined to recommend acceptance.

Specifically, I ask that:
1. parts of the paper be made more formal, with proper preliminaries.
2. Better discussion and explanations concerning the sensitivity property.

**Strengths And Weaknesses:**

Strengths:

- The problem is relevant and interesting, and the authors provide a novel perspective.
- The parts of the paper which are supposed to convey intuition are well-written and convincing.
- The experimental evidence is compelling.
- The results are interesting and suggest new directions of research. Specifically, the results suggest that one may wish to construct a tight relaxation with few discontinues and low sensitivity. Alternatively, proving that such a tight relaxation cannot exist is also interesting.

Weaknesses:


- The paper misses some key definitions. To make the paper friendlier to the casual reader, it would be helpful to add definitions for convex relaxations, tightness of relaxation, certified robustness, etc. Currently, those terms are defined in a very vague, non-quantitative, and intuitive manner. Just an example (but certainly not the only one), Theorem 1 refers to $CR(\theta,r)$, the certified robustness, which is not defined anywhere in the paper, at least not where I could see.

- The experimental evidence is not supported by theory, and the contribution itself is somewhat limited. The paper does make an interesting observation, but it would have been more interesting to follow up on this observation.

- I found the explanation and intuition about the sensitivity property (Section 4.3) to be too vague.
1. First, just looking at the degree of rational function seems to be too coarse. It suggests that the sensitivity of $\frac{1}{1+x^2}$ should be treated the same as $1+x^2$, which is not very meaningful. Also, depending on the domain, the coefficients of the polynomial should also matter. Usually, sensitivity is measured with respect to Lipschitz constants or modulus of continuity; why is it justified to only look at the degree?

2. Second, I would appreciate a discussion about why sensitivity is measured with respect to changes in the first layer. While the importance of the first layer is intuitively clear in this context, I think there is room for discussion in the scope of the paper.

3. Finally, I found it a bit hard to track the calculation of sensitivity for the different models. Adding a section to the appendix where those calculations are carefully derived would be a good idea.

---

> ### Author Response · Authors · 2022-08-06
> **Response to Reviewer BLjF**
>
> We thank the reviewer for their useful suggestions which we have incorporated in our new revision. In the following we provide answers to raised concerns.
>
>
> **Q1: Can you make the background more formal (i.e., add definitions of convex relaxations, tightness, certified robustness)?** \
> We have improved the background by moving definitions of key terms from other parts of the paper to Section 2, and further expanding this section with missing definitions and intuitive explanations as suggested by the reviewer. We hope this makes the paper more approachable, and are happy to work on further improving the preliminaries.
>
>
> **Q2: Is the experimental evidence supported by theory? Could you follow up on your observations?** \
> Yes, we believe so. In terms of theoretical contributions, we have Theorem 2, with a non-trivial proof in Appendix C.2, which states that IBP and DeepZ are provably continuous with respect to network parameters. For other relaxations, we have counterexamples showing cases when they are discontinuous. Regarding sensitivity, we have derived provable upper bounds on sensitivity for each relaxation in Section 4.3. As a followup, we systematically experimented with modifying existing relaxations in Section 6. Our conclusion was that improving unfavorable relaxations is hard and does not directly lead to state-of-the-art results.
>
>
> **Q3: For sensitivity, why is it justified to consider only the degree of the polynomial, and not coefficients? Why not measure the Lipschitz constant?** \
> As hinted in the paper, looking only at the degree does not provide all of the information, and coefficients certainly matter as well. However, coefficients and the Lipschitz constant are influenced by the weights, making it difficult to compute the worst-case bound in the closed form. Thus, we chose to focus on the polynomial degree as a metric that depends only on the number of layers in the network and the number of neurons per layer. More importantly, this formulation is appropriate for our goal of showing that some relaxations are much more sensitive than IBP, allowing us to gain insight into the paradox of certified training. In our new revision, we have expanded the corresponding discussion in Section 4.3.
>
>
> **Q4: Why is sensitivity defined only with respect to the first layer?** \
> We used this simplification to make the calculation of worst case bounds tractable. While other layers certainly matter for the optimization, including them in the calculation would make it more difficult to obtain closed form bounds. Further, while this would change the exact value of the bounds, it would not affect our conclusions regarding the relative sensitivity of relaxations compared to IBP. We have included a corresponding explanation in the new revision.
>
>
> **Q5: Can you provide more detailed derivations of sensitivity?** \
> We thank the reviewer for the suggestion. In our latest revision, we include an extended version of these derivations in newly added Appendix J, and reference this from the main text in Section 4.3.

---

> > ### Comment · Reviewer_BLjF · 2022-08-28
> >
> > Thank you for the revised version.
> > The revision addresses most of the points I've raised.
> > I think the paper can be a good fit to TMLR and so I've recommended to accept.

---

### Review · Reviewer_bUar · 2022-07-28

**Summary Of Contributions:**

The paper studies the phenomenon that certified training methods of the (ReLU-network) convex relaxation type may have better robustness performance despite worse tightness. The paper identifies two further properties, (dis)continuity (of the loss landscape) and sensitivity (w.r.t. perturbation of the first layer’s weight), which are argued via extensive experiments to influence the optimization and thereby the performance. In particular, the paper points out that a method which has worse tightness can have better continuity and sensitivity properties, leading to better performance, though the paper also demonstrates that the interdependency among these properties is complex.

**Requested Changes:**

Aside from addressing my criticisms:
- Can the paper clarify why it makes sense to conclude, say, CROWN-IBP (R) is more sensitive than IBP, based on the estimate $O(M^{B+1})$? These estimates seem worst-case, and clearly $R_m(\delta) \subseteq R_n(\delta)$ for any $m \leq n$. There is still the possibility that the effective sensitivity that the gradient descent trajectory experiences under CROWN-IBP (R) is much less than $O(M^{B+1})$.
- Can the paper clarify why the only difference accounted for hBox, in the comparison with IBP on Page 7, is the affine layers? Doesn’t it make sense that hBox, a discontinuous method, should have very high sensitivity?
- Can the paper specify the confidence interval / error margin for Table 4, if possible? Some of these numbers are too close to make a clear comparison.

**Strengths And Weaknesses:**

Robust training is a large topic and the question being pursued by the paper is definitely of interests to wide audience. The extensive experimentation over several common methods is valuable to the literature. I particularly find the parameterization in Table 4, as a way to probe understanding, quite interesting.

My main reservation is that the two identified properties are argued to be key influencing factors to robustness mainly by the comparison between IBP vs other methods (Table 2 and relevant discussions). What about other pairs? For example, CROWN-IBP (R) has worse tightness (Table 1) than CROWN, is discontinuous like CROWN, and shares the same order of sensitivity as CROWN, but Table 2 shows it consistently beats CROWN. Can the hypothesized properties predict this outcome? A similar observation for CROWN-IBP (R) vs DeepZ.

The paper proposes that worse continuity and sensitivity tend to impede optimization as an explanation, but is it clear that the loss region where discontinuity or high sensitivity occurs is responsible for the bad final performance? Or is it that methods that have worse continuity / sensitivity in this paper actually have an overall loss landscape that is unfavorably non-convex to gradient descent training? Recall that the loss landscape is inherently non-convex, regardless of tightness, continuity, sensitivity. In other words, what the paper has shown is that continuity and sensitivity can be used to qualitatively predict if a method may offer more robust performance, but it is unclear from the paper if continuity and sensitivity have a causal relation with performance.

There are a few weaknesses of the specific measures of continuity and sensitivity:
- Continuity does not seem quantifiable, so one can only compare continuous vs discontinuous methods.
- Sensitivity is defined w.r.t. the first layer only. Why shouldn’t other layers matter to the optimization?

---

> ### Author Response · Authors · 2022-08-06
> **Response to Reviewer bUar**
>
> We thank the reviewer for providing helpful feedback and raising several interesting questions. The answers to raised concerns follow.
>
> **Q1: Could continuity and sensitivity be used to compare any pair of relaxations (e.g., CROWN-IBP (R) and CROWN)?** \
> Yes, continuity and sensitivity can be generalized from binary to numerical metrics to compare any pair of relaxations. For continuity, this can be done by empirically measuring the number of discontinuities along one direction in the loss landscape as we have done in Figure 5 and Figure 6 (left). These experiments show that CROWN indeed has more discontinuities than CROWN-IBP (R). Similarly, sensitivity could be generalized by considering the exact upper bound degree (instead of only whether it is greater than 1), as derived in Section 4.3. This would explain why DeepZ, which has the sensitivity of $O(3^B M^{B+1})$, performs worse than CROWN-IBP (R) which has the sensitivity of $O(M^{B+1})$. The reason why we binarized these properties is because we focus on the important problem of explaining why widely used IBP relaxation, which is the least tight, performs better than all of the other relaxations when used in certified training. We included a discussion along these lines in newly added Appendix I.
>
> **Q2: Is it clear that the loss region where discontinuity or high sensitivity occurs is responsible for bad performance or is it that methods that have worse continuity/sensitivity actually have an overall loss landscape that is unfavorably non-convex to gradient descent training?** \
> This is an interesting question, which was partly the motivation for the experiments we performed in Section 6. Concretely, to establish the causal link between continuity/sensitivity and the final performance, one would have to change the continuity or sensitivity of a relaxation, keeping everything else constant, and then measure the new performance. However, it turned out that for most of the relaxations in our systematic investigation changing one property also affects another property. Thus, establishing a causal link is quite hard, but our intuition is that these properties indeed negatively affect the optimization as sketched in Figure 4. Moreover, while proving that there is a causal relationship would definitely be interesting, we believe that our findings advance the understanding of certified training and are useful for practitioners when deciding which relaxation to utilize in training.
>
>
> **Q3: Why is sensitivity defined only with respect to the first layer?** \
> We used this simplification to make the calculation of worst case bounds tractable. While other layers certainly matter for the optimization, including them in the calculation would make it more difficult to obtain closed form bounds. Further, while this would change the exact value of the bounds, it would not affect our conclusions regarding the relative sensitivity of relaxations compared to IBP. We have included a corresponding explanation in the new revision.
>
>
> **Q4: Does it make sense to conclude that CROWN-IBP (R) is more sensitive than IBP given that its effective sensitivity could be much smaller than the upper bound of $O(M^{B+1})$?** \
> Yes, we believe that it is reasonable to conclude that CROWN-IBP (R) is more sensitive than IBP. IBP provably has the sensitivity of exactly 1, while CROWN-IBP (R) has the sensitivity between 1 and $O(M^{B+1})$, meaning that the unlikely best possible value of sensitivity of CROWN-IBP (R) is still not better than the sensitivity of IBP.
>
>
> **Q5: Should hBox, as a discontinuous method, also have very high sensitivity?** \
> No, this is not the case as sensitivity is a local property. Concretely, there could be many discontinuous pieces of the loss landscape, and each piece could have arbitrary sensitivity. Thus, continuity and sensitivity, as defined in this paper, are completely independent properties.
>
>
> **Q6: Can you expand Table 4 to include error margins?** \
> Yes. We have added standard deviations of certified robustness after 4 independent runs with different random seeds to Table 4. Further, guided by the suggestion of the reviewer and the observation that, in our experimental setup, indeed for very small $\omega$ it is sometimes hard to make robust statements regarding the relative order of two observed relaxations, we have expanded this experiment by investigating several larger values of $\omega$, and 4 additional relaxations, i.e., different variants of LooseIBP-DC. This lead to more detailed observations regarding the tradeoffs between tightness and continuity, reaffirming and further strengthening the conclusions reached in this experiment. We refer the reviewer to our expanded discussion at the bottom of Section 6 for more details.

---

> > ### Comment · Reviewer_bUar · 2022-08-10
> > **Reply**
> >
> > Thanks a lot for the detailed response and for updating the manuscript. It definitely looks better. I have one suggestion (though feel free to argue against it). Continuity, per Section 4.2, is a binary property, but certain analyses concerning this property have to expand it to count discontinuities on the landscape or on the optimization trajectory, i.e. are more post hoc than predictive and thus rather dependent on the experimental setup. This could be remarked in a suitable way in Section 4.2. In light of this, I view continuity, though not a perfect measure, a good attempt that can be developed further in subsequent works.

---

> > > ### Author Response · Authors · 2022-08-11
> > > **Response**
> > >
> > > Thank you for the suggestion. We have now updated the paper with this discussion at the end of the first paragraph in Section 4.2. Please let us know if you have any further comments that could improve the presentation.

---

### Review · Reviewer_nPi7 · 2022-07-28

**Summary Of Contributions:**

The proposed work is concerned with certified robustness, in particular certified training of neural networks. In this approach to mitigating the sensitivity of neural networks, an upper and lower bound of the output change effected by a bounded change of the input.
The resulting bounds are optimized during training, allowing to optimize models that achieve a high degree of *certifiable* robustness.

The starting point of the proposed work is the "paradox of certified training", an observation in the literature that using tighter bounds in the certification of robustness does not usually improve the results. This should not be possible for globally optimally trained neural networks, suggesting that optimization difficulties are at the heart of the issue.

To illucidate this mechanism, the present work studies two related properties of certificates that impact optimization. The discontinuity of the resulting loss function and it's sensitivity to small perturbations of the network parameter.
The authors proceed to measure these quantities for multiple certificates and observe that indeed the tighter certificates produce more irregular and/or sensitive loss functions than that obtained by interval bound propagation (IBP), a simple application of interval arithmetic.
This explains the fact that the IBP outperforms all other approaches in terms of certified robustness, even though the latter are based on tighter certificates.

**Broader Impact Concerns:**

I do not have any concerns about the broader impact of this work.

**Requested Changes:**

I do not request any changes.

**Strengths And Weaknesses:**

The proposed work is well written and motivated. It seems to thoroughly study the properties of different approaches in certified robustness  and provides a plausible mechanism to explain the "paradox of certified training".

I am not an expert in the literature on certified training but based on my reading, I recommend the paper to be accepted.

---

> ### Author Response · Authors · 2022-08-06
> **Response to Reviewer nPi7**
>
> We thank the reviewer for positive comments on our work and a favorable recommendation.

---

### Author Response · Authors · 2022-08-06
**General Response**

We thank all reviewers for their feedback and are pleased that they believe our work is relevant and well-motivated (Rev. nPi7 and BLjF), provides a novel perspective (Rev. BLjF), and is a valuable addition to the literature (Rev. bUar). Guided by suggestions of reviewers, we have uploaded a new revision of our paper, clearly marking the additions compared to the original submission. We answer concerns raised by reviewers in individual responses below each review, and are happy to continue the discussion further.

---

### Decision · Action_Editors · 2022-09-15

**Recommendation:** Accept as is

**Comment:**

After one round of discussion with the reviews, the reviewers concurred that the authors addressed their concerns, and the paper is in a much better shape. With that in mind, the paper will be a good contribution to TMLR.